# FLARE: Faithful Logic-Aided Reasoning and Exploration

## Abstract

Modern Question Answering (QA) and Reasoning approaches based on Large Language Models (LLMs) commonly use prompting techniques, such as Chain-of-Thought (CoT), assuming the resulting generation will have a more granular exploration and reasoning over the question space and scope. However, such methods struggle with generating outputs that are faithful to the intermediate chain of reasoning produced by the model. On the other end of the spectrum, neuro-symbolic methods such as Faithful CoT (F-CoT) and Logic-LM propose to combine LLMs with external symbolic solvers. While such approaches boast a high degree of faithfulness, they usually require a model trained for code generation and struggle with tasks that are ambiguous or hard to formalise strictly. We introduce **F**aithful **L**ogic-**A**ided **R**easoning and **E**xploration (**FLARE**), a novel interpretable approach for traversing the problem space using task decompositions. We use the LLM to plan a solution, formalise the query into facts and predicates, which form the problem space, using a logic programming code and simulate that code execution using an exhaustive multi-hop search over the defined space. Our method allows us to compute the faithfulness of the reasoning process w.r.t. the generated code and explicitly trace the steps of the multi-hop search without relying on external solvers. Our methods achieve SOTA results on **7** out of **9** diverse reasoning benchmarks. We also show that model faithfulness positively correlates with overall performance and further demonstrate that **FLARE** allows pinpointing the decisive factors sufficient for and leading to the correct answer with optimal reasoning during the multi-hop search. Our findings reveal that successful traces exhibit, on average, a $18.1\%$ increase in unique emergent facts, a $8.6\%$ higher overlap between code-defined and execution-trace relations, and a $3.6\%$ reduction in unused code relations.

## 1 Introduction

Complex Reasoning in natural Question Answering (QA) tasks assumes the capability to explore the problem space of the designated query with a formalised set of facts, relations, commonsense knowledge and logical implications. In line with this, LLMs have been enhanced with CoT (Wei et al., 2022) prompting, which supplements the QA process by generating intermediate reasoning chains given a set of in-context examples (Brown et al., 2020a), as shown in Figure 1. This allowed for advancement in commonsense (Madaan et al., 2022), symbolic (Wang et al., 2022; Sprague et al., 2024) and mathematical (Jie et al., 2023) reasoning. Although CoT allows for a problem exploration in natural language steps, such an approach has been shown to cause performance degradation for reasoning tasks involving multi-step planning (Valmeekam et al., 2022; Suzgun et al., 2023), problem exploration (Yao et al., 2022), and arithmetic tasks (Hendrycks et al., 2021b; Madaan & Yazdanbakhsh, 2022b). These discrepancies arise as CoT suffers from a limited ability to decompose, search, verify and backtrack using intermediate rationale chains (Yao et al., 2022), cascading hallucinations and errors (Ling et al., 2023) and that natural language might not be an optimal representation for describing the reasoning process (Li et al., 2024). Simultaneously, LLM output has been shown to be unfaithful and inconsistent w.r.t. the intermediate CoT rationale (Jacovi et al., 2024; Lanham et al., 2023b; Turpin et al., 2023).

To mitigate the problem of CoT faithfulness and allow for more robust reasoning during QA, Lyu et al. (2023, Faithful CoT) and Logic-LM (Pan et al., 2023) suggested generating code which is

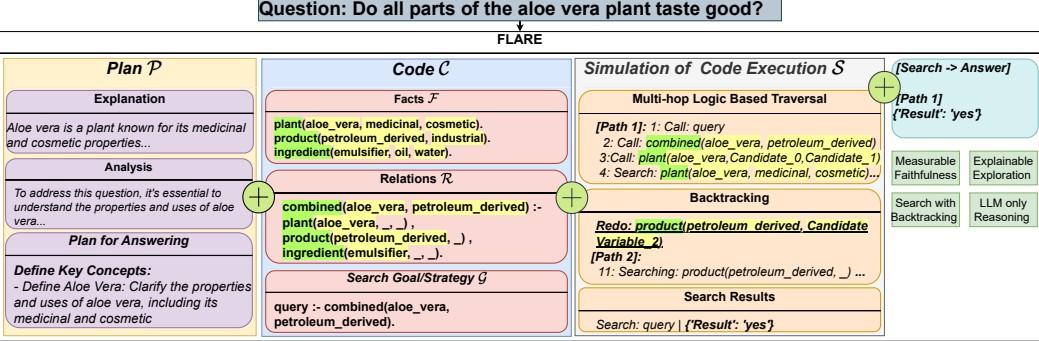

Figure 1: A depiction of the *plan*, *code* and simulated *search* in FLARE. Each module is generated separately and iteratively, allowing us to obtain the final answer. The green and yellow highlighted text shows the overlap between the facts and the relations between the code and the simulated search.

further executed using an external symbolic solver. Producing and executing code enables the generation of outputs guided by external solvers, leveraging search with backtracking to explore the problem space effectively. However, strict translations of natural language queries into code, such as *autoformalisation* (Szegedy, 2020; Wang et al., 2018), is a non-trivial task involving direct inference of implicit commonsense and domain-specific knowledge and the ability to align abstract and informal concepts directly to constrained formal definitions for further execution (Wu et al., 2022). An example query, *"Do all parts of the aloe vera plant taste good?"*, is challenging to formalize or address with a strict algorithmic solution, as it requires interpretative, deductive and context-dependent reasoning, referred to as soft or fuzzy reasoning. Using external solvers makes such fuzzy reasoning impossible and requires consistently generating syntactically correct executable code. While some LLMs have coding capabilities stemming from their pretraining (Jiang et al., 2024; Aryabumi et al., 2024), relative code consistency is more probable with models explicitly trained for coding (Chen et al., 2021).

To overcome these problems, we propose Faithful Logic-Aided Reasoning and Exploration (FLARE), an interpretable method that allows for planning, fuzzy reasoning, and traversing the problem space with backtracking, exact task decomposition, and measuring faithfulness. In FLARE, given a natural language query, we prompt an LLM to sequentially generate a *plan* that includes an analysis and the logical steps necessary for formalising and answering the question, a logic programming (Wielemaker et al., 2012) *code* that allows formalising the query into a set of facts, relations and their composition forming the space for exploring that query and the *search*, which is an LLM-generated code execution simulation. An illustration of FLARE can be seen in Figure 1. In our framework, the generated code must not be consistently executable by an external solver, allowing for the soft-formalisation of natural language. Although we see that even generalist LLMs are able to produce executable code in $\geq 50\%$ of cases. FLARE allows us to measure the faithfulness of the outcome w.r.t. the simulated code execution by directly comparing the search paths produced by the external solver to that LLM generation. This comparison also allows for pinpointing model hallucinations and inconsistencies. We systematically study the effectiveness of our method using 4 general-purpose LLMs of varying scales across 9 diverse QA and 3 logical inference benchmarks, covering Math World Problems, Multi-hop QA, Relation inference, deductive and analytical reasoning and show that our method achieves state-of-the-art results in 7 out of 9 QA datasets and 2 out of 3 logic datasets in comparison to CoT, F-CoT and Logic-LM. We also show that the method is competitive for models tuned for coding, with an average overall increase of $16\%$ over F-Cot and $9\%$ over CoT. Our findings show that model accuracy strongly correlates with the faithfulness of the reasoning process towards search traces from the simulated code execution. We also provide ablations showing that the model can interpretably pinpoint hallucinations, underutilized knowledge, and the limitations of the search over the problem space. Our key contributions are the following:

- We introduce FLARE a novel paradigm for logic-aided and interpretable formalisation and search over the problem space in QA and logic reasoning tasks.

- We perform a systematic evaluation across 9 QA and 3 logical inference benchmarks and 4 models of varying scales, showing the advantages of using FLARE for QA in a few-shot setup over prior approaches.

- The modularity of FLARE allows defining a simple ingrained method for measuring model faithfulness, which is further shown to be strongly correlated with performance.

- We further show that using FLARE allows us to interpretably and rigorously detect hallucinations along with sub-optimal and inconsistent reasoning patterns.

## 2 RELATED WORK

**Reasoning in Natural Language**  Few-shot prompting (Brown et al., 2020b) has been shown to be an effective approach for increasing the reasoning capabilities of LLMs in natural language generation (Gehrmann et al., 2021; Reif et al., 2022; Sanh et al., 2022). LLM reasoning can be further enhanced with prompting techniques such as CoT (Wei et al., 2022), which attempts to segment reasoning into explicitly written intermediate steps. Concurrent work has also proposed that models *"think step by step"* (Kojima et al., 2022), or divide the problem into subtasks before the solution (Zhou et al., 2023, Least-to-Most). These approaches have been shown to suffer from arithmetic inaccuracies (Lewkowycz et al., 2022; Hendrycks et al., 2021a) and reasoning inconsistencies (Madaan & Yazdanbakhsh, 2022a). Further attempts have been made to add a planning stage before reasoning by dividing the process into recursive plan formulation and execution steps (Yao et al., 2023b; Wang et al., 2023a). The *plan* generation step in FLARE is a hybrid technique inspired by these methods but focused on generating a natural language strategy for formalising the query into code.

**Reasoning with Search**  Several lines of work propose using techniques to expand the reasoning paths over the problem space. Self-consistency decoding (Wang et al., 2023b) is an approach used to sample many natural language reasoning paths and take a majority vote for an answer. Another popular approach is Tree-of-Thoughts (ToT; Yao et al., 2023a), which proposes to explore the query with reasoning similar to a tree traversal, where each state is created and evaluated using an LLM. Similar techniques try to adapt symbolic search approaches akin to DFS, BFS (Besta et al., 2024), A* (Lehnert et al., 2024) or other combinations (Gandhi et al., 2024) with direct tuning (Lehnert et al., 2024), imitation training (Yang et al., 2022) or few-shot prompting (Zhang et al., 2024). It must be noted that all of these techniques have only been tested in constrained mathematical puzzle-solving and algorithmic domains like the 24 Game (Yang et al., 2022), Countdown (Wikipedia, 2024), Sorting (Besta et al., 2024), maze solving (Yang et al., 2022), Sokoban (Lehnert et al., 2024), and others. Although the *search* component of FLARE has some similarities to these techniques, we argue that our method allows for generalistic reasoning with interpretable multi-hop search through simulated code execution.

**Reasoning with Formalisation**  Another line of research has tried formalising natural language queries into code (Gao et al., 2023; Li et al., 2024) or pseudo-code (Chae et al., 2024; Gandhi et al., 2024). This allows the translation of the query into a strict structure and offloads the reasoning and search components to deterministic solvers like Python Chen et al. (2023), PDDL Lyu et al. (2023); Liu et al. (2023), DataLog Lyu et al. (2023) and others. While models are capable of synthesising programs (Austin et al., 2021; Nijkamp et al., 2023) and benefit from the use of code in numerical and algorithmic reasoning settings (Chen et al., 2023; Gao et al., 2023), the usage of code for general QA has not been rigorously explored. The reasons are that formalisation from natural language into a strict and executable code is challenging (Wu et al., 2022), following the exact syntactic constraints of the programming language not abundantly used during pre-training is onerous (Liu et al., 2024) and can require models explicitly tuned for coding (Chen et al., 2021). Using an external solver for reasoning also limits the capability for soft reasoning in commonsense knowledge and implications. Although we formalise the natural language query into a logic programming Prolog program during the *code* generation part of FLARE, we do not explicitly require the code to be executable and do not use external solvers during inference. This allows for the further use of the LLM for soft-reasoning to simulate code execution in a logic-based problem space traversal similar to Prolog while circumventing the need for code tuning a generalist model.

| Method | Math Word Problems | | | | | Multi-hop QA | | | Relation |
|---|---|---|---|---|---|---|---|---|---|
| | GSM8K | SVAMP | MultiArith | ASDiv | AQuA | StrategyQA | Date | Sport | CLUTRR |
| Llama-3.1-8B$_{FLARE}$ | **72.7** | **86.0** | **96.3** | **83.1** | **62.9** | **70.2** | **59.3** | 76.6 | 36.8 |
| Llama-3.1-8B$_{F\text{-}CoT}$ | *0* | *0* | *0* | *0* | *12.2* | 53.2 | *0* | *0* | 32 |
| Llama-3.1-8B$_{CoT}$ | 59.2 | 58.6 | 60.1 | 61.9 | 35 | *2.9* | 20.9 | **95.8** | **42.2** |
| CmDR$_{FLARE}$ | **52.4** | **74.0** | **84.5** | **72.2** | **43.7** | **67.0** | **52.3** | **78.9** | 29.1 |
| CmDR$_{F\text{-}CoT}$ | *0* | *0* | *0* | *0* | *0* | 59.7 | *0* | *0* | *8.6* |
| CmDR$_{CoT}$ | 46.5 | 57.3 | 83.1 | 37.2 | 28.3 | *21.3* | 47.4 | 55.2 | 29.5 |
| CmDR+$_{FLARE}$ | **71.4** | **83.5** | **90.4** | **81.3** | **55.9** | **70.8** | 61.8 | 77.7 | 41.0 |
| CmDR+$_{F\text{-}CoT}$ | *0* | *0* | *0* | *0* | *15.4* | 57.6 | *0* | *0* | 35.3 |
| CmDR+$_{CoT}$ | 48.7 | 81.1 | 86.6 | 44.6 | 44.1 | *48.4* | **79.1** | 62.6 | 42.5 |
| GPT-3.5$_{FLARE}$ | *68.1* | 82.7 | **98.3** | **85.4** | 55.1 | **65.5** | **82.4** | 85.6 | **49.8** |
| GPT-3.5$_{F\text{-}CoT}$ | 75.8 | **83.0** | 95.3 | 81.7 | *53.5* | 51.5 | 73.5 | 52.3 | 12.1 |
| GPT-3.5$_{CoT}$ | **79.8** | *82.4* | 98.2 | 75.8 | 59.4 | *51.7* | 69.9 | **95.8** | *4.3* |

Table 1: The following table shows the performance of each of the tested models given a technique for reasoning. Each **bold**, underlined, and *italicised* element highlights the best, second best and worst technique per specific model. The overall best method per dataset is highlighted in green .

**Reasoning Faithfulness**  An explanation is considered *faithful* if it explicitly and accurately describes the reasoning process of the model during inference (Gilpin et al., 2018; Jacovi & Goldberg, 2020). In the context of prompting techniques such as CoT, we are interested in the faithfulness of the intermediate reasoning chains towards the final output. Faithful intermediate reasoning chains should not just look *plausible* (Herman, 2017) but have exact reflections of the problem exploration and reasoning used to arrive at the final answer. Natural language reasoning chains prevalent in CoT and similar methods are shown to be unfaithful, either masking the reasoning biases (Turpin et al., 2023) of the model or outright ignoring the intermediate reasoning (Lanham et al., 2023a). In FLARE, we introduce a method to seamlessly measure the faithfulness of the final outcome w.r.t. completed search.

## 3 METHODOLOGY

### 3.1 LLM SIMULATED SEARCH

FLARE comprises three modules for generating a *plan*, *code* and simulated *search* for answering a natural language query $\mathcal{Q} = \{T_1^{\mathcal{Q}} \ldots T_{|\mathcal{Q}|}^{\mathcal{Q}}\}$, where each $T_i^{\mathcal{Q}}$ is a token in the query $\mathcal{Q}$.

**Generating A Plan**  For each query $\mathcal{Q}$, given an LLM $\mathcal{M}$, we initially use instructions $\mathcal{I}^{\mathcal{P}}$ to prompt it to generate a *plan* $\mathcal{P}$, which should be comprised of task explanation, analysis and a plan for further formalising the query. An example of this can be seen in the *plan* section in Figure 1. We use in-context few shot examples $\mathcal{E}_{\mathcal{P}}$ of such *plan* generations along with **greedy** decoding for obtaining the final plan.

$$\mathcal{P}_i \sim \arg\max p_{\mathcal{M}}(T_i^{\mathcal{P}} \mid T_{:i-1}^{\mathcal{P}}, \mathcal{E}_{\mathcal{P}}, \mathcal{Q}, \mathcal{I}^{\mathcal{P}}) \tag{1}$$

Where $\mathcal{P}_i$ and $T_i^{\mathcal{P}}$ is the $i$-th token in the generated *plan* $\mathcal{P}$ and $p_{\mathcal{M}}$ is the probability of the next token over the vocabulary obtained from model $\mathcal{M}$.

**Generating Code**  After generating the *plan*, we use instructions $\mathcal{I}^{\mathcal{C}}$ to prompt the LLM $\mathcal{M}$ to generate a Prolog code $\mathcal{C}$, an example of which can be seen in Figure 1. We append executable code generation samples $\mathcal{C}_{\text{sample}}$ to the previous in-context examples $\mathcal{E}_{\mathcal{P}}$ and obtain few-shot code generation demonstrations $\mathcal{E}_{\mathcal{C}} = [\mathcal{E}_{\mathcal{P}}; \mathcal{C}_{\text{sample}}]$

$$\mathcal{C}_i \sim \arg\max p_{\mathcal{M}}(T_i^{\mathcal{C}} \mid T_{:i-1}^{\mathcal{C}} \mathcal{E}_{\mathcal{C}}, \mathcal{Q}, \mathcal{I}^{\mathcal{P}}, \mathcal{P}, \mathcal{I}^{\mathcal{C}}) \tag{2}$$

$$\mathcal{F}_{code}, \mathcal{R}_{code}, \mathcal{G}_{code} = \text{EXTRACT}(\mathcal{C}_i)$$

Where $\mathcal{C}_i$ and $T_i^{\mathcal{C}}$ is the $i$-th token in the generated *code* $\mathcal{C}$.

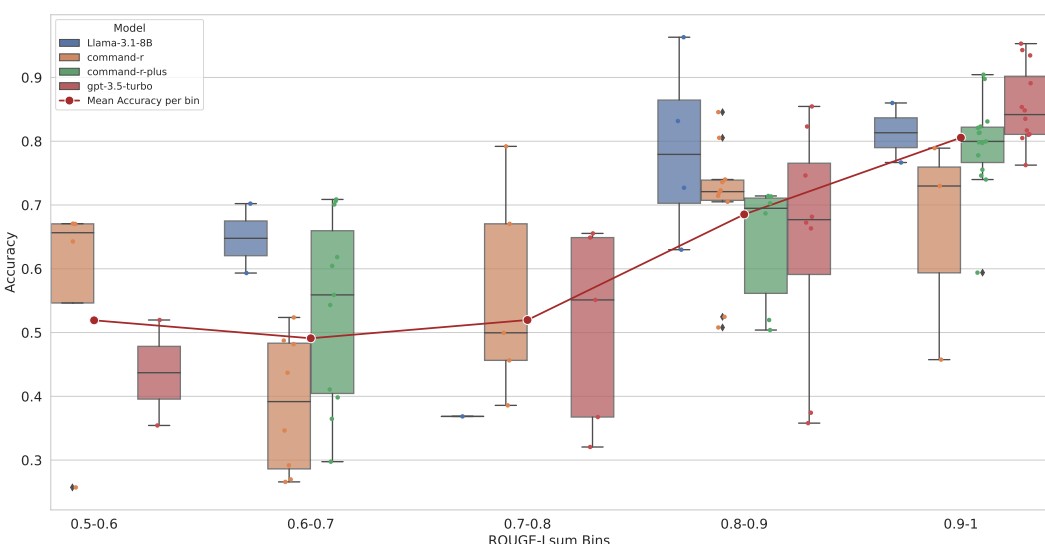

Figure 2: The trend of mean model accuracy w.r.t mean faithfulness for all the models.

**Benefits of Prolog** Prolog is a symbolic logic-programming engine (Bowen, 1979) designed for heuristic search over Horn Clauses (Chandra & Harel, 1985). As a declarative programming paradigm (Lloyd, 1994), the code is expressed as the logic of computation, expressed as a set of facts $\mathcal{F}$ and relations $\mathcal{R}$ defining the problem space, with the goal $\mathcal{G}$ being a first-order logic combination of them. Prolog employs depth-first search (DFS) (Bowen, 1979) for sub-goal decomposition and traversal of the problem space, satisfying $\mathcal{G}$ through a sequence of steps known as the *trace*. Each step either confirms/invalidates a sub-goal, expands the search tree, or retries failed sub-goals with new combinations. An example of such a search is shown in Figure 1. Prolog supports exhaustive search by exploring all paths that satisfy or fail the goal. This explicit segmentation of facts, relations, and search strategies simplifies query formalization. As a declarative language, Prolog enables segmentation using a simple regexp heuristic, referred to as EXTRACT in Equation (2) and Equation (3). Including exhaustive traces in-context allows an LLM to simulate sub-goal decomposition, backtracking, and intermediate goal invalidation, discussed further in the next paragraph.

**Simulating Search** After generating the logic-programming *code*, we want to simulate program execution by generating a problem space traversal trace with our LLM $\mathcal{M}$. We use instructions $\mathcal{I}^{\mathcal{S}}$ and update our in-context samples by appending search traces $\mathcal{S}_{\text{sample}}$ constructed from Prolog execution of sample codes $\mathcal{C}_{\text{sample}}$, i.e. $\mathcal{E}_{\mathcal{S}} = [\mathcal{E}_{\mathcal{C}}; \mathcal{S}_{\text{sample}}]$.

$$\mathcal{S}_i \sim \arg\max p_{\mathcal{M}}(T_i^{\mathcal{S}} \mid T_{:i-1}^{\mathcal{S}} \mathcal{E}_{\mathcal{C}}, \mathcal{Q}, \mathcal{I}^{\mathcal{P}}, \mathcal{P}, \mathcal{I}^{\mathcal{C}}, \mathcal{C}, \mathcal{I}^{\mathcal{S}}) \qquad (3)$$

$$\mathcal{A}_{search}, \mathcal{F}_{search}, \mathcal{R}_{search} = \text{EXTRACT}(\mathcal{S}_i)$$

Where $T_i^{\mathcal{S}}$ is the $i$-th token in the generated *search* trace $\mathcal{S}$. During iterative problem space traversal, we can segment the facts $\mathcal{F}_{search}$, relations $\mathcal{R}_{search}$, completed and backtracked paths with their answers $\mathcal{A}_{search}$ used during the search simulation. To get the final answer we update in-context samples with their correct final answers $\mathcal{A}_{\text{sample}}$ from the executed search $\mathcal{S}_{\text{sample}}$, $\mathcal{E}_{\mathcal{A}} = [\mathcal{E}_{\mathcal{S}}; \mathcal{A}_{\text{sample}}]$ and use instructions $\mathcal{I}^{\mathcal{A}}$ to obtain the final answer from the model.

$$\mathcal{A}_{Final} \sim \arg\max p_{\mathcal{M}}(T_i^{\mathcal{A}} \mid T_{:i-1}^{\mathcal{A}} \mathcal{E}_{\mathcal{C}}, \mathcal{Q}, \mathcal{I}^{\mathcal{P}}, \mathcal{P}, \mathcal{I}^{\mathcal{C}}, \mathcal{C}, \mathcal{I}^{\mathcal{S}}, \mathcal{S}, \mathcal{I}^{\mathcal{A}}) \qquad (4)$$

The prompts used for generating each part in FLARE can be seen in Appendix A along with a complete example in Table 8 and a pseudo-code in Appendix A.3.

### 3.2 DETECTING REASONING INCONSISTENCIES

For each query $\mathcal{Q}$ given the *code* $\mathcal{C}$ and the simulated *search* $\mathcal{S}$ along with the extracted facts $\mathcal{F}_{code}, \mathcal{F}_{search}$ and relations $\mathcal{R}_{code}, \mathcal{R}_{search}$ from each designated module, we aim to detect the inconsistencies during the reasoning process of the LLM. We use exact string matching between all

these facts and relations in *code* and simulated *search*.

$$\forall i, \exists j \quad \text{such that} \quad \mathcal{F}^i_{\text{code}} = \mathcal{F}^j_{\text{search}} \quad \text{and} \quad \forall v, \exists q \quad \mathcal{R}^v_{\text{code}} = \mathcal{R}^q_{\text{search}} \tag{5}$$

$$\forall j, \exists i \quad \text{such that} \quad \mathcal{F}^i_{\text{code}} = \mathcal{F}^j_{\text{search}} \quad \text{and} \quad \forall q, \exists v \quad \mathcal{R}^v_{\text{code}} = \mathcal{R}^q_{\text{search}} \tag{6}$$

With this framework in mind, we define two reasoning failure modes. In the *first* failure mode, given that some fact or relation was used in the simulated *search* but did not exist in the generated *code*, i.e. $\exists j$ such that $\mathcal{F}^j_{\text{search}} \notin \mathcal{F}_{\text{code}}$, we claim that the LLM has *hallucinated*. We postulate that the model either produced incomplete knowledge during formalisation to *code* or created a piece of non-existing information during the *search*. We do not consider facts that emerged during a direct inference step within the simulated search during our calculation. For example, if

| Dataset | ChatGPT (gpt-3.5-turbo) | | | |
| --- | --- | --- | --- | --- |
| | Standard | CoT | Logic-LM | FLARE |
| PrOntoQA | 47.40 | 67.80 | 61.00 | **73.40** |
| LogicalDeduction | 40.00 | 42.33 | **65.67** | 58.60 |
| AR-LSAT | 20.34 | 17.31 | 26.41 | **27.39** |

Table 2: Comparison of Direct Prompting, CoT, Logic-LM and FLARE.

we are dealing with a mathematical query $4 \cdot (5 + 6) = ?$, the search would involve separately evaluating the expression $5 + 6 = 11$. In this case, 11 will not be treated as a hallucinated fact within the search but rather as an emergent fact obtained from direct inference. The *second* failure mode is the reciprocal case, where a fact or relation present in the *code* is not used during the *search*. We refer to this phenomenon as *sub-optimal reasoning* as it shows that the LLM could not explore the problem space completely or injected unsuitable knowledge during formalisation into *code*.

## 3.3 MEASURING FAITHFULNESS

We propose a method to measure the faithfulness of the LLM reasoning process when using FLARE. As mentioned in Section 3.1, for each query in a dataset $\mathcal{D} = [\mathcal{Q}_1, \ldots, \mathcal{Q}_{|\mathcal{D}|}]$, we generate a set of codes $\Phi = [\mathcal{C}_1, \ldots, \mathcal{C}_{|\Phi|}]$ and simulated problem space searches $\Psi = [\mathcal{S}_1, \ldots, \mathcal{S}_{|\Psi|}]$. We use the Prolog engine to execute all of the codes $\Phi$ and obtain a set of correctly written programs $\Phi'$ and exact search paths $\Psi'$. As we do not require explicit programmatic correctness during inference in FLARE for any code $\mathcal{C}_i$, some Prolog executions resulting in an error are filtered out in $\Psi'$. To assess model reasoning faithfulness towards code formalisations, we compare the search paths $\Phi'$ obtained from Prolog execution with their designated counterparts $\Phi'_{gen}$ generated by the LLM from the same code. We use ROUGE (Lin, 2004) to compute the matching score for each executed and simulated search path. In particular, we use ROUGE-Lsum, which uses the longest common subsequence (LCS) over each line to obtain the final score. This method fits our cause as a line in a Prolog search execution represents a single logic step within the traversal. This allows us to measure the similarity of the reasoning contents and structure in exact and simulated searches.

## 4 EXPERIMENTAL SETUP

### 4.1 DATASETS

To evaluate FLARE, we use a benchmark of 9 tasks covering Math Word Problems (MWP), multi-hop QA and relation inference, and 3 common logical reasoning datasets. For testing numeric and mathematical reasoning, we follow CoT (Wei et al., 2022) by including GSM8K (Cobbe et al., 2021), SVAMP (Patel et al., 2021), MultiArith (Roy & Roth, 2015), ASDiv (Miao et al., 2020) and AQuA (Ling et al., 2017). Among these, GSM8K, SVAMP, MultiArith and ASDiv cover elementary and middle school arithmetic word problems with a set of integers or decimals as the answer. AQuA is a multiple-choice numerical, symbolic reasoning dataset where each answer is a mathematical expression containing notations, values and expressions not defined in the query. We also test FLARE using three multi-hop QA datasets. We use StrategyQA (Geva et al., 2021), which is a boolean QA task that requires sub-goal decomposition and a multi-hop reasoning strategy to answer. The example *"Do all parts of the aloe vera plant taste good?"* used in Figure 1, is taken from StrategyQA. The multi-hop QA testing also includes Date and Sports Understanding, subsets of BIG-Bench (bench authors, 2023). The tasks involve inferring an exact date given some calculations in the relative time period and understanding if an artificially created sports statement is feasible. Furthermore, we assess FLARE on Relational Inference using CLUTRR (Sinha et al., 2019), which involves inferring

| Method | CmDR$_{FLARE}$ | CmDR$_{plan-only}$ | CmDR+$_{FLARE}$ | CmDR+$_{plan-only}$ | GPT-3.5$_{FLARE}$ | GPT-3.5$_{plan-only}$ |
|---|---|---|---|---|---|---|
| GSM8K | **52.4** | 24.7 | **71.4** | 40.7 | **68.1** | 36.1 |
| AQuA | **43.7** | 35.0 | **55.9** | 55.1 | **55.1** | 54.3 |
| StrategyQA | **67.0** | 65.5 | 70.8 | **75.7** | **65.5** | 62.3 |

Table 3: The table shows the accuracy of an LLM with FLARE compared to prompting for a final answer directly after generating (plan-only) a plan $\mathcal{P}$.

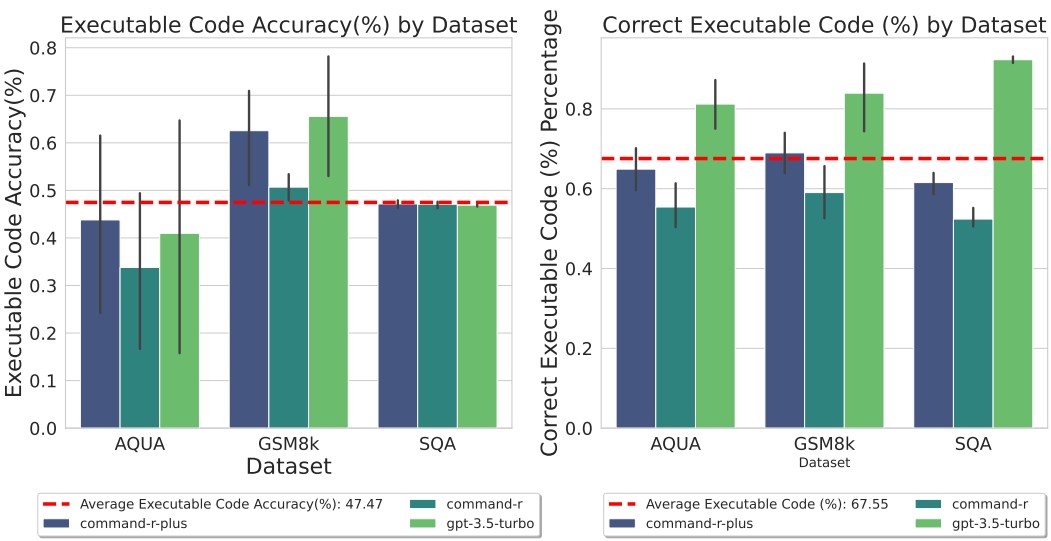

Figure 3: The figure shows the percentage of executable code per model (right) and the accuracy of the executable code when answering the queries (left).

the familial relation between two entities mentioned in a natural language description of the partial family graph. We evaluate FLARE on challenging logic inference datasets: ProntoQA (Saparov & He, 2023), AR-LSAT (Zhong et al., 2021), and LogicalDeductions from BigBench (et al., 2023), focusing on harder subsets proposed by Pan et al. (2023). These datasets, covering deductive, analytical, and logical reasoning, allow us to assess FLARE's performance. Details, including descriptions and examples, are in Table 9 of Appendix A.

## 4.2 BENCHMARKS

We compare FLARE with CoT (Wei et al., 2022) as a prompting method that reasons using natural language chains and with F-CoT (Lyu et al., 2023) and Logic-LM (Pan et al., 2023) that formalise the query into a code and offload the reasoning to an external symbolic solver. We use Llama3.1 (8B) (Dubey et al., 2024), CmDR (30B) (Cohere, 2024), CmDR+ (100B) (Cohere, 2024) and GPT3.5 (Brown et al., 2020b) ($\geq 100B$ (Ye et al., 2023)). As the coding model OpenAI Codex (code-DaVinci-002) (Chen et al., 2021) used in F-CoT has been deprecated, we replace it with the new GPT3.5 as suggested by OpenAI and recalculate the results accordingly.

## 5 RESULTS

### 5.1 FEW-SHOT PROMPTING

To evaluate FLARE, we use a set of models of varying sizes on diverse benchmarks, as defined in Section 4. We compare the performance of each model while using FLARE, CoT and F-CoT prompting. The results for F-CoT and CoT on all the models are computed using the codebase of the original study (Lyu et al., 2023). We additionally compare Logic-LM and FLARE using the logic reasoning benchmarks proposed in Pan et al. (2023).

| Model | Avg. Number of Paths | Avg. #Hops per path | Avg. #Fails per path | Avg. Total Hops | Avg. Total Fails |
|---|---|---|---|---|---|
| | | **Incorrect Answers** | | | |
| Llama-3.1-8B$_{FLARE}$ | 1.55 | 11.12 | 1.52 | 15.09 | 2.26 |
| CmDR$_{FLARE}$ | 1.51 | 6.55 | 0.68 | 10.56 | 1.39 |
| CmDR+$_{FLARE}$ | 0.92 | 7.52 | 1.13 | 8.57 | 1.32 |
| GPT-3.5 | 0.68 | 5.22 | 0.71 | 5.32 | 0.74 |
| | | **Correct Answers** | | | |
| Llama-3.1-8B$_{FLARE}$ | 1.43 | 9.12 | 0.62 | 12.36 | 0.96 |
| CmDR$_{FLARE}$ | 1.19 | 7.10 | 0.42 | 11.29 | 0.66 |
| CmDR+$_{FLARE}$ | 0.97 | 7.19 | 0.42 | 8.22 | 0.61 |
| GPT-3.5$_{FLARE}$ | 0.82 | 5.65 | 0.26 | 5.69 | 0.27 |

Table 4: The table depicts the difference in the average explored paths, hops, and fails during the reasoning process, which leads to incorrect or correct answers. The purple colour illustrates that incorrect reasoning paths have fewer explorations that led to Failed search paths.

**LLMs for general reasoning**   Our results, presented in Table 1, show that using FLARE allows the LLMs to achieve state-of-the-art results on 7 out of 9 datasets, with an average $28\%$ increase over CoT. We can see a clear trend that FLARE increases the performance compared to CoT and F-CoT for all the models of varying scales. We also see that LLMs that are not explicitly tuned for coding suffer massive degeneracies when using F-CoT. We postulate that they are unable to consistently produce executable programs that satisfy a predefined scheme in F-CoT, thus resulting in an error during execution. This further highlights the value of simulating program execution using an LLM instead of using external solvers. The results show that using FLARE yields more benefit on datasets that require longer chains of multi-hop and symbolic reasoning, like AQuA and StrategyQA. Our findings in Table 2 show that FLAREachieves state-of-the-art results on 2 out of 3 logic inference benchmarks with $10\%$ increase over CoT and  $7\%$ increase over Logic-LM.

**LLMs for code generation**   To understand the effect of FLARE on models tuned for coding, we use GPT3.5 (Brown et al., 2020a) as it was the OpenAI suggested succession model for Codex (Chen et al., 2021) which is used in F-CoT and possesses strong coding capabilities (Ye et al., 2023). The results in Table 1 show that using FLARE is beneficial for models that are tuned for coding and boost the accuracy with a $16\%$ increase over F-CoT and $9\%$ over CoT. The reason is that many natural language queries with non-trivial formalisations are more suited to be tackled with more commonsense soft reasoning than direct code execution. This is evident in Table 1 where FLARE and CoT are consistently better than F-CoT in StrategyQA, Sports and CLUTRR. The opposite case of numeric and algorithmic heavy reasoning tasks is also covered by FLARE as it maintains strong performance similar to F-CoT on MWP problems Table 1. Consequently, FLARE allows combining algorithmic formalisation with simulated soft-reasoning, circumventing the pitfalls of using a deterministic external solver while still producing a query formalisation and problem space traversal.

## 5.2   IS SIMULATING SEARCH USEFUL?

To understand if simulating a search over the problem space is useful, we compare the performance of FLARE where we only generate the *plan* without the subsequent *code* or *search* components. We refer to this framework setup as *plan-only*, which can be seen in Figure 1 if we were to use only the *plan* for answer generation. We completed this ablation using CmDR, CmDR+, and GPT-3.5, and we used GSM8K, AQuA, and StrategyQA as our baselines. The results in Table 3 confirm that all of the models suffer massive performance degradation from $61.1 \rightarrow 49.9$ when omitting the *code* and the *search* components of FLARE. We hypothesise that this is caused by insufficient problem space exploration when using the *plan-only* setting. Furthermore, we have already seen in Table 1 that in methods, like F-CoT, that do not use simulated problem space exploration for soft-reasoning and only rely on *plan* and *code*, the performance also deteriorates even resulting in a complete breakdown of reasoning over the designated datasets. This can be viewed as a constrained version of FLARE with *code-only* execution. Consequently, our results show that simulating problem space traversal is highly beneficial as it avoids the pitfalls posed by *plan-only* and *code-only* modes by exploring the problem space more rigorously and soft-reasoning during that traversal instead of using external solvers.

## 5.3 Faithful Reasoning Improves Performance

As described in Section 3, using FLARE allows us to measure the faithfulness of the LLM reasoning process by comparing the simulated problem space traversals $\Phi'_{gen}$ with actual traces $\Phi'$ produced from a symbolic Prolog solver. To do this, we initially compute the percentage of syntactically correct executable code each LLM produces. We can see from the right part of Figure 3 that all of the models are capable of producing correct executable Prolog code in $67\%$ of cases on average and $\geq 50\%$ of cases at the very least. This shows that the simulated searches $\Phi'_{gen}$ can be considered a representative sample that will be further used to accurately measure the faithfulness of the simulated search w.r.t. the generated code. After measuring the reasoning faithfulness for each model, we want to understand what impact it has on the performance of the LLM. In Figure 2, we segment the models w.r.t. their ROUGE-Lsum scores. The results show that model perfor-

| Model | UEF (%) in Search | OR (%) | UR (%) in code |
|---|---|---|---|
| **Correct Answers** | | | |
| Llama-3.1-8B$_{\text{FLARE}}$ | 74.14 | 43.65 | 5.73 |
| CmDR$_{\text{FLARE}}$ | 59.06 | 35.96 | 4.02 |
| CmDR+$_{\text{FLARE}}$ | 64.30 | 34.47 | 4.54 |
| GPT-3.5$_{\text{FLARE}}$ | 64.46 | 37.55 | 1.90 |
| **Avg. (Correct)** | 65.49 | 37.91 | 4.05 |
| **Incorrect Answers** | | | |
| Llama-3.1-8B$_{\text{FLARE}}$ | 54.69 | 35.04 | 9.28 |
| CmDR$_{\text{FLARE}}$ | 54.50 | 32.76 | 6.23 |
| CmDR+$_{\text{FLARE}}$ | 44.12 | 24.98 | 8.22 |
| GPT-3.5$_{\text{FLARE}}$ | 36.02 | 24.44 | 6.94 |
| **Avg. (Incorrect)** | 47.33 | 29.31 | 7.67 |
| $\Delta$ | 18.16 | 8.60 | -3.62 |

Table 5: The table shows how the percentage of unique emergent facts (UEF) in search, overlapping relations (OR) between code and search, and unused relations (UR) in code impact answer correctness.

mance is strongly positively correlated with reasoning faithfulness. However, we also observe in the left part of Figure 3 that executing semantically precise code results in an accurate answer only in $47\%$ of cases on average. Indeed, having a simulated search trace with a ROUGE-Lsum faithfulness score of 1, would be equivalent to simply executing the program as proposed in F-CoT. Yet we have priorly shown that F-CoT struggles with reasoning tasks that are hard to formalise and require multi-hop commonsense and soft reasoning. These two discoveries show that optimal LLM reasoning, conditioned on a search in the problem space, should be increasingly faithful toward the facts, relations and the search strategy defined within the code while simultaneously maintaining the capability for soft-reasoning along more abstractly defined concepts. Our results show that FLARE allows LLMs to maintain a similar reasoning capacity.

## 5.4 What is important during the search?

We expand the analysis of the simulated search traces to detect the reasons which can lead to optimal reasoning within an LLM. For this purpose, we calculate several statistics, like the average number of explored paths, average and total hops and failures per path, for each model during the simulated traversal. The failure in a path is an invalidation of a solution for a sub-goal explored during the search, which is used for backtracking, as explained in Section 3. Calculating these statistics is simple as the *search* component of FLARE, seen in Figure 1, is a structured simulation of a Prolog trace, where each line contains a hop of reasoning inference. We split these statistics for the

| Model | Hops per Paths | Halluc. (%) | Unutil. (%) |
|---|---|---|---|
| Llama-3.1-8B | 9.4 | 63.3 | 62.9 |
| CmDR | 6.7 | 54.7 | 56.9 |
| CmDR+ | 7.2 | 54.3 | 56.3 |
| GPT-3.5 | 5.5 | 49.3 | 52.1 |

Table 6: The table shows the changes in simulated search averaged statistics when using FLARE w.r.t model scale from 8B to 100B+. Hallucinations (Halluc.) refer to facts and predicates only used in trace, while unutilised knowledge (Unutil.) relates to the facts and relations only seen in the code.

reasoning paths that lead to correct or incorrect outcomes. Our results in Table 4 show that LLM performance and reasoning optimally are not directly connected to the amount of explored paths or multi-hop inferences per path. We also see that traces that lead to incorrect answers have a higher number of failures per path and in total. We explain this phenomenon with the hypothesis that LLMs with traces that were optimal for reasoning and led to correct answers could skip exploring degenerate solutions due to strong commonsense reasoning capabilities. Further analyses focus on identifying inconsistencies and failure modes (Section 3.2). By comparing relations in code with those in search traces, we measure emergent hallucinations and unused relations, highlighting areas

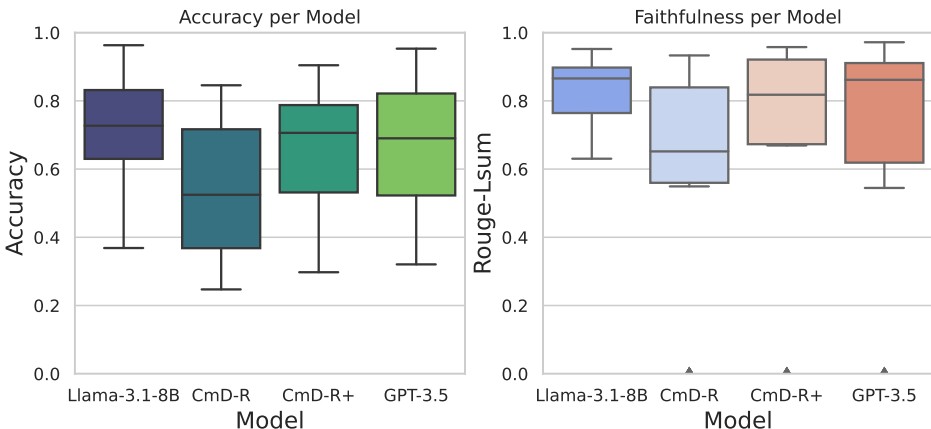

Figure 4: The effect of the model parameter scale from 8B to 100B+ on model accuracy (left) and faithfulness (right).

of sub-optimal reasoning. Additionally, we assess the uniqueness of emergent facts per inference hop, which indicates the extent of problem-space exploration (Table 5). The results in Table 5 show consistently over each model that, on average, traces that lead to correct answers had a higher percentage of unique emergent facts (UEF) and overlap in the relations (OR) used between the code and search, while the portion of underutilized relations (UR) was lower. This means that optimal reasoning with an LLM requires a great degree of problem-space exploration with fewer relation hallucinations during the search and more relation utilization from the defined code. This aligns with our prior discoveries, which show a strong correlation between simulated search faithfulness towards the formalised code and model performance. Our framework FLARE has these reasoning patterns ingrained within its inference pipeline.

## 5.5 THE EFFECT OF SCALE

We want to assess the impact of the number of parameters in the model on the overall performance and faithfulness. The results in Figure 4 show no precise relation between model scale, performance and faithfulness. However, scaled models from the same family, i.e. CmDR (30B) and CmDR+ (100B), show improvements in reasoning faithfulness and model performance. We can also see in Table 6 that as the model size increases, the average number of hops and the portion of hallucinations and unutilised knowledge decreases. This further confirms our prior assumptions that models with strong commonsense soft-reasoning capabilities can skip steps during the search while maintaining the knowledge and structure of the traversal strategy outlined in the code.

## 6 CONCLUSION

This work introduces FLARE, a novel approach for logic-aided interpretable formalisation and reasoning with simulated search over the problem space. We show that models of varying scales obtain state-of-the-art results compared to prompting paradigms like CoT and F-CoT. We further pinpoint that using FLARE allows us to perform soft-reasoning with simulated search, making it flexible for diverse reasoning benchmarks. We introduce a method to measure model reasoning faithfulness w.r.t. the problem formalization ingrained within FLARE. Our results show that model performance is positively correlated with the faithfulness of the reasoning process. The systematic studies of the method show the benefits of using simulated search compared to natural language reasoning and external symbolic solvers. We further show that using FLARE allows us to interpretably and rigorously detect hallucinations and sub-optimal and inconsistent reasoning patterns.

REPRODUCIBILITY REPORT

To reproduce the results of our study, we provide the complete codebase, processing pipelines and prompts for each dataset. The only model hyper-parameter we explicitly fix is the temperature for greedy decoding. We also make the inference of all of the models using FLARE, F-CoT and CoT across all of the datasets publicly available for further experimentation and exploration.

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

# A APPENDIX

## A.1 LLM PROMPTS

We define straight-forward prompts for generating *plan*, *code* and *search* simulation in FLARE, which can be observed in Appendix A.2.

## A.2 DATASET STATISTICS

The datasets used in this study encompass a variety of domains, specifically targeting the performance of the models in interpreting Math Word Problems, multi-hop question answering, and relational inference. Table 9 provides a detailed breakdown of each dataset, including the number of few-shot in-context samples (shots), the number of test samples, and representative examples from each dataset. The datasets provide a comprehensive basis for evaluating the models' abilities to handle complex tasks across different domains, facilitating an in-depth analysis of model performance under few-shot conditions.

| Task | Prompt | Description |
|---|---|---|
| **Plan Generation** | Generate an explanation and analysis, and plan to generate a prompt for writing a swi-prolog code for the last task. The 3 sections should be exactly outlined. Your plan should show enough intermediate reasoning steps towards the answer. Construct the plan as much as you can and describe the logic specifically. When constructing the plan for the code prompt, actively use swi prolog search capabilities. | Detailed instructions for generating an outline and plan, with an emphasis on reasoning steps and using Prolog's search capabilities. |
| **Code Generation** | Write a Prolog code to solve using the plan. If there are unknown or stochastic atoms or predicates, fill in the values for them as a logical assumption and add a comment in the same line Assumed atom/predicate". Do not use write and read commands within the code. The code should be very detailed and utilize swi prolog capabilities to the fullest. To run the program, at the end create a predicate named "query" that returns the correct numerical answer. The last line of the program should be the commented-out driver predicate "query". Write only the code. | Instructions for generating a Prolog code based on the plan with assumptions for unknown atoms. Emphasizes code details and a final "query" predicate. |
| **Simulated Search** | Ignoring the read commands, explicitly write out the search paths that are explored by the code: #### Here are the paths [Starting Search Simulation]: #### [Path 1]: | A task to simulate and display the search paths that the Prolog code would follow during execution. |
| **Final Answer** | Given the plan, the code and the explored search paths answer the question above. Answer with the correct numerical answer. ##### Here is the answer: | Final prompt asking for the correct numerical answer based on the previous steps. |

Table 7: Table of Prompts for Plan, Code, Simulated Search, and Final Answer generation for GSM8K (Cobbe et al., 2021).

## A.3 FLARE PSEUDO-CODE

Below, we present the pseudo-code for the execution of the *plan*, *code*, and *search* procedures in FLARE. The pseudo-code describes the modular pipeline in FLARE for tackling natural language queries with faithful simulated search.

- **Plan Generation:** This stage creates a structured natural language outline of the reasoning process, breaking down the query into logical steps and analysis. The plan serves as the foundation for formalization into a logic-based representation.

- **Code Generation:** Based on the generated plan, a logic programming code (e.g., in Prolog) is synthesized. This code formalizes the query into a set of facts, relations, and goals, which collectively define the problem space for reasoning.

- **Search Simulation:** The generated code is utilized to simulate a search trace over the problem space. This includes iterative reasoning, backtracking when goals are unmet, and extracting emergent facts or relations during the process.

Each of these stages is implemented as a modular component. The generation from each of the stages feeds into the next, allowing seamless integration and incremental improvement in reasoning accuracy. A detailed pseudo-code is provided below in Appendix A.3.

---

**Algorithm 1** FLARE Methodology: Faithful Logic-Aided Reasoning and Exploration

---

**Require:** Query $\mathcal{Q}$, Language Model $\mathcal{M}$
**Ensure:** Answer $\mathcal{A}$
 1: **Initialization:** Load few-shot examples for plans ($\mathcal{E}_P$), code ($\mathcal{E}_C$), and search traces ($\mathcal{E}_S$)
 2: **Input:** Natural language query $\mathcal{Q}$
 3: **procedure** GENERATE PLAN
 4:     Prompt $\mathcal{M}$ with instructions $\mathcal{I}_P$ and examples $\mathcal{E}_P$ to generate a plan $\mathcal{P}$
 5:     $\mathcal{P} \leftarrow \arg\max\ p_{\mathcal{M}}(T_P|T_{P:<i}, \mathcal{E}_P, \mathcal{Q}, \mathcal{I}_P)$
 6: **end procedure**
 7: **procedure** GENERATE CODE
 8:     Append examples $\mathcal{E}_C$ to $\mathcal{E}_P$
 9:     Prompt $\mathcal{M}$ with instructions $\mathcal{I}_C$ to generate logic programming code $\mathcal{C}$
10:     $\mathcal{C} \leftarrow \arg\max\ p_{\mathcal{M}}(T_C|T_{C:<i}, \mathcal{E}_C, \mathcal{Q}, \mathcal{P}, \mathcal{I}_C)$
11:     $(F_{\text{code}}, R_{\text{code}}, G_{\text{code}}) \leftarrow \text{EXTRACT}(\mathcal{C})$
12: **end procedure**
13: **procedure** SIMULATE SEARCH
14:     Append search trace examples $\mathcal{E}_S$ to $\mathcal{E}_C$
15:     Prompt $\mathcal{M}$ with instructions $\mathcal{I}_S$ to simulate a search trace $\mathcal{S}$
16:     $\mathcal{S} \leftarrow \arg\max\ p_{\mathcal{M}}(T_S|T_{S:<i}, \mathcal{E}_S, \mathcal{Q}, \mathcal{P}, \mathcal{C}, \mathcal{I}_S)$
17:     $(F_{\text{search}}, R_{\text{search}}, \mathcal{A}_{\text{search}}) \leftarrow \text{EXTRACT}(\mathcal{S})$
18:     **while** Goal $\mathcal{G}_{\text{code}}$ is not satisfied **do**
19:         Explore next sub-goal in $\mathcal{S}$
20:         **if** Sub-goal fails **then**
21:             Backtrack to the previous state (Learned through in-context sampels)
22:         **end if**
23:     **end while**
24: **end procedure**
25: **procedure** FINAL ANSWER GENERATION
26:     Append correct answers from $\mathcal{A}_{\text{search}}$ to examples
27:     Prompt $\mathcal{M}$ with instructions $\mathcal{I}_A$ to finalize answer $\mathcal{A}$
28:     $\mathcal{A} \leftarrow \arg\max\ p_{\mathcal{M}}(T_A|T_{A:<i}, \mathcal{E}_A, \mathcal{Q}, \mathcal{P}, \mathcal{C}, \mathcal{S}, \mathcal{I}_A)$
29: **end procedure**
30: **return** $\mathcal{A}$

---

| Section | Content |
|---------|---------|
| Question | "Can Reiki be stored in a bottle?" |
| Explanation | Reiki is a form of alternative therapy that involves the channeling of energy to promote healing and relaxation. It is often administered through the practitioner's hands, with the energy flowing directly to the recipient. The question raises the possibility of storing Reiki energy in a bottle, which may be a common query due to the abstract nature of energy healing. |
| Analysis | To address the question of whether Reiki can be stored in a bottle, it is important to consider the nature of Reiki energy, the principles of energy healing, and the practical implications of containing and preserving such energy. |
| Plan | • Understand Reiki Energy: ...

• Explore Energy Storage: ...

• Consider Symbolism and Intention: ...

• Evaluate Practicality: ...

• Reflect on Tradition: ... |
| Code | % Define the nature of Reiki energy reiki_energy(spiritual_energy).
reiki_energy(channeling_through_touch).
% Define the feasibility of storing energy in a bottle
store_energy_in_bottle(feasible) :-
reiki_energy(spiritual_energy),
reiki_energy(channeling_through_touch).
% Define the assumption that storing Reiki energy in a bottle is not a common practice not_common_practice(store_reiki_in_bottle).
% Query to determine if Reiki can be stored in a bottle
query :-
store_energy_in_bottle(feasible),
not_common_practice(store_reiki_in_bottle). |
| Search Paths | Path 1:   – 0: Start of execution: Beginning Search
    – 1: Call: query
    – 2: Call: store_energy_in_bottle(feasible)
    – 3: Call: reiki_energy(spiritual_energy)
    – 4: Call: reiki_energy(channeling_through_touch)
    – 5: Call: not_common_practice(store_reiki_in_bottle)
    – 6: Call: feasible=no
    – 7: Fail: feasible=no — 'Result': 'Search Failed'
    – 8: Fail: query — 'Result': 'Search Failed' |
| Answer | "No, Reiki energy cannot be stored in a bottle based on the logical evaluation of its abstract, non-physical nature and traditional practices of Reiki." |

Table 8: Complete example of FLARE

| Domain | Dataset | Shots | Test Samples | Example |
|--------|---------|-------|--------------|---------|
| Math Word Problems | GSM8K | 8 | 1,319 | Q: A robe takes 2 bolts of blue fiber and half that much white fiber.
How many bolts in total does it take?
A: 3 |
| | SVAMP | 8 | 1,000 | Q: Dan had $3 left with him after he bought a candy bar. If he had $4 at the start, how much did the candy bar cost?A: 1 |
| | MultiArith | 8 | 600 | Q: A pet store had 13 siamese cats and 5 house cats. During a sale they sold 10 cats.
How many cats do they have left?
A: 8 |
| | ASDiv | 8 | 2,096 | Q: Adam has five more apples than Jackie. Jackie has nine apples. How many apples does Adam have?
A: 14 |
| | AQuA | 8 | 254 | Q: A man walks at 5 kmph for 6 hrs and at 4 kmph for 12 hrs. His average speed is
Answer option: A)4 1/3 km/h, B)7 2/3 km/h, C)9 ½ km/h, D)8 km/h, E)81 km/h
A: A |
| Multi-hop QA | StrategyQA | 6 | 2,290 | Q: Did Aristotle use a laptop?
A: False |
| | Date Understanding | 10 | 359 | Q: Yesterday was April 30, 2021. What is the date tomorrow in MM/DD/YYYY?
A: "05/02/2021" |
| | Sports Understanding | 10 | 977 | Q: Is the following sentence plausible? Lionel Messi was called for icing?
A: False |
| Relational Inference | CLUTRR | 8 | 1,042 | Q: [Carlos] is [Clarence]'s brother. [Carlos] and his sister, [Annie], went shopping.
asked her mom [Valerie] if she wanted anything, but [Valerie] said no.
How is [Valerie] related to [Clarence]?
A: "mother" |

Table 9: The statistics and examples of the datasets used in benchmarking. Shots refers to the number of few-shot in-context samples used during benchmarking.

