# OpenReview forum: "FLARE: Faithful Logic-Aided Reasoning and Exploration"
_ICLR.cc/2025/Conference — Submitted to ICLR 2025_

### Official Review · Reviewer_KpGf · 2024-10-27

**Soundness:** 3
**Presentation:** 3
**Contribution:** 3
**Rating:** 8
**Confidence:** 4

**Summary:**

Authors introduce a new logic aided interpretable formalization search propmting mechanism namely FLARE, where given a query a plan is generated by the LLM following which a logical prolog code is generated given the plan which is simulated with multihop search.

**Strengths:**

1) The paper is generally is well written with crisply defined mathematical notations and easy to follow presentation of authors works
2) Experimentation section of the paper is sound demonstrating the effectiveness of the proposed method

**Weaknesses:**

1) Expensive: Given that the LLMs have been used for planning, code generation and the execution of code via multi-hop search, instead of relying on symbolic solvers, this might increase the cost for computation.
2) Weak baselines: Experiments have been performed with relatively simpler datasets. Authors should experiment with more complex datasets that are already present in the literature for question answering such as LogiQA [1], PrOntoQA [2],  ProofWriter [3]
3) Figures: Figure 1 present in the paper is too cluttered and hard to read. Please break it down so that it is easy to read or please add a more detailed explanation in the appendix
4) Prompt Templates: Ideally since the proposed method is a prompting method, prompt templates are necessary to evaluate the performance of the proposed method in detail but Appending A2 does not include and since there is no supplementary material provided with paper, it is hard to effectively judge the paper in total.

[1] LogiQA: A Challenge Dataset for Machine Reading Comprehension with Logical Reasoning
[2] Language Models Are Greedy Reasoners: A Systematic Formal Analysis of Chain-of-Thought
[3] ProofWriter: Generating Implications, Proofs, and Abductive Statements over Natural Language

**Questions:**

1) How cost effective is the proposed method over baselines methods since a lot of search is needed?
2) How effective is the proposed method over the methods like LLM + external symbolic solver such as LogicLM [1] ?
3) As mentioned in the cons section, can you please show the effectiveness of proposed method on more complex logical reasoning datasets?

[1] Logic-LM: Empowering Large Language Models with Symbolic Solvers for Faithful Logical Reasoning

---

> ### Author Response · Authors · 2024-11-19
> **Rebuttal for the Reviewer KpGf**
>
> We thank the reviewer for their feedback on our work and proceed to sequentially address the raised Weaknesses and Questions:
>
> General comments on Weaknesses:
>
> ```Expensive: Given that the LLMs have been used for planning, code generation and the execution of code via multi-hop search, instead of relying on symbolic solvers, this might increase the cost for computation.```
>
> The speed of the external solvers used for baseline methods (F-CoT) is highly dependent and sensitive towards the executing system and choice of the setup that runs it (F-CoT can be run with different versions of Python, PDDL etc). This makes a direct comparison with LLM inference impossible. However, per the reviewer's request, we compare the number of tokens and the inference speed for generating CoT vs F-CoT (without the external solver added time) vs FLARE.
>
> | model          |   ('plan', 'mean') |   ('plan', 'std') |   ('code', 'mean') |   ('code', 'std') |   ('trace', 'mean') |   ('trace', 'std') |   ('trace_percentage', 'mean') |   ('trace_percentage', 'std') |
> |:---------------|-------------------:|------------------:|-------------------:|------------------:|--------------------:|-------------------:|-------------------------------:|------------------------------:|
> | Llama-3.1-8B   |            552.544 |           146.769 |            241.274 |           128.231 |             336.255 |            281.753 |                        26.9211 |                       11.3238 |
> | command-r      |            511.435 |           160.666 |            180.228 |           132.454 |             266.82  |            268.343 |                        23.0468 |                       15.1715 |
> | command-r-plus |            577.032 |           184.77  |            189.203 |           151.211 |             223.105 |            209.69  |                        19.7404 |                       12.5113 |
> | gpt-3.5-turbo  |            557.774 |           161.733 |            177.533 |           128.255 |             164.41  |            184.529 |                        16.3258 |                       10.686  |
>
> Our results show that the total length of the trace is on average composed of ~250 tokens which corresponds to an approximately 20% increase compared to CoT and F-CoT. Despite this marginal increase, the impact on overall inference speed is negligible w.r.t. to Cot and F-CoT. In particular, the additional tokens are not a reason for a computational overhead as the computational cost of adding a small number of tokens (such as the additional ~250 tokens in FLARE traces) is minimal compared to the overall cost of processing the base sequence. This is specifically true when the sequence lengths remain well below the model's maximum context window.
>
> ```Weak baselines: Experiments have been performed with relatively simpler datasets. Authors should experiment with more complex datasets that are already present in the literature for question answering such as LogiQA [1], PrOntoQA [2], ProofWriter [3]```
>
> As clarified in the paper, we used the benchmarks from Faithful-CoT paper, which are also widely present across the literature. Per the reviewers request, we add a direct comparison with to Logic-LM on the hardest sets of ProntoQA[2], AR-LSAT[3] and Logical Deduction [4]. These datasets are used in the original study of Logic-LM and are considered challenging logic inference benchmarks. We opted not to include PrOntoQA and ProofWriter, as they exhibit considerable overlap with the benchmarks already used in our study (e.g., CLUTTER, AQUA, and StrategyQA). Instead, we prioritised datasets that present unique challenges to demonstrate the efficacy of FLARE better.
>
> | Dataset           | Standard | CoT   | Logic-LM       | FLARE          |
> |-------------------|----------|-------|----------------|----------------|
> | **PrOntoQA**      | 47.40    | 67.80 | 61.00          | **73.40**      |
> | **LogicalDeduction** | 40.00    | 42.33 | **65.67**      | 58.60          |
> | **AR-LSAT**       | 20.34    | 17.31 | 26.41          | **27.39**      |
>
>
> Our findings show that FLARE achieves state-of-the-art results on $2$ out of $3$ logic inference benchmarks with $10$ percent increase over CoT and $~7$ percent increase over Logic-LM. These results highlight FLARE’s ability to handle challenging reasoning tasks compared to existing approaches.

---

> > ### Author Response · Authors · 2024-11-19
> > **Continuation of Rebuttal for the Reviewer KpGf**
> >
> > ```Figures: Figure 1 present in the paper is too cluttered and hard to read. Please break it down so that it is easy to read or please add a more detailed explanation in the appendix```
> >
> > We have further improved Figure 1 by updating the layout and adding further annotations for clarity. The updated figure includes a direct demonstration of the whole search process with backtracking. We have also added a complete example of FLARE in the Appendix of the paper.
> >
> >
> > ```Prompt Templates: Ideally since the proposed method is a prompting method, prompt templates are necessary to evaluate the performance of the proposed method in detail but Appending A2 does not include and since there is no supplementary material provided with paper, it is hard to effectively judge the paper in total.```
> >
> > We have added all of the implementation details, all of the used prompts and results in the submitted updated paper, supplementary materials and the code. All of these can also be found in our anonymously shared code repository: https://anonymous.4open.science/r/FLARE-D88B/README.md
> >
> >
> >
> > **General comments on Questions:**
> >
> >
> > ```How cost effective is the proposed method over baselines methods since a lot of search is needed?```
> >
> > As outlined in the previous comments on weaknesses, uur results show that the total length of the trace is on average composed of ~250 tokens which corresponds to an approximately 20% increase compared to CoT and F-CoT. Despite this marginal increase, the impact on overall inference speed is negligible w.r.t. to Cot and F-CoT. In particular, the additional tokens are not a reason for a computational overhead as the computational cost of adding a small number of tokens (such as the additional ~250 tokens in FLARE traces) is minimal compared to the overall cost of processing the base sequence. This is specifically true when the sequence lengths remain well below the model's maximum context window.
> >
> > We also have provided the overall statistics for the completed search in Table 4 within the paper. Our results show that while the models are capable of performing 11 multi-hop steps within a single DFS branch/path, usually only 1-3 paths are explored for a total of 250 tokens on average. We see a similar distribution from the traces generated from actual code execution. We postulate that most of the queries do not require the depth of the search to be high and also show that models use fuzzy reasoning to skip failing paths and reasoning lines (Tables 4, 5,6), thus truncating the need for exhaustive search.
> >
> >
> >
> > ```How effective is the proposed method over the methods like LLM + external symbolic solver such as LogicLM [1] ? As mentioned in the cons section, can you please show the effectiveness of proposed method on more complex logical reasoning datasets? ```
> >
> > As previously clarified, Faithful-CoT is also a method like LLM + symbolic solver, and we directly compare FLARE with the same baselines.
> >
> > Per the reviewers request, we add a direct comparison with to Logic-LM on the hardest sets of ProntoQA[2], AR-LSAT[3] and Logical Deduction [4]. These datasets are used in the original study of Logic-LM and are considered challenging logic inference benchmarks. We opted not to include PrOntoQA and ProofWriter, as they exhibit considerable overlap with the benchmarks already used in our study (e.g., CLUTTER, AQUA, and StrategyQA). Instead, we prioritised datasets that present unique challenges to demonstrate the efficacy of FLARE better.
> >
> > | Dataset           | Standard | CoT   | Logic-LM       | FLARE          |
> > |-------------------|----------|-------|----------------|----------------|
> > | **PrOntoQA**      | 47.40    | 67.80 | 61.00          | **73.40**      |
> > | **LogicalDeduction** | 40.00    | 42.33 | **65.67**      | 58.60          |
> > | **AR-LSAT**       | 20.34    | 17.31 | 26.41          | **27.39**      |
> >
> >
> > Our findings show that FLARE achieves state-of-the-art results on $2$ out of $3$ logic inference benchmarks with $10\%$ increase over CoT and $~7\%$ increase over Logic-LM. These results highlight FLARE’s ability to handle challenging reasoning tasks compared to existing approaches.
> >
> > We respectfully encourage the reviewer to revisit the scores in light of the clarifications, improvements and additional experiments provided in our rebuttal. If any concerns remain, we would greatly appreciate additional feedback or suggestions for improvement.

---

> ### Comment · Reviewer_KpGf · 2024-11-19
>
> I appreciate the detailed rebuttals by the authors but I would like to seek some clarifications.
> 1) Search Process:
> Please correct me if I have misunderstood something.
> Is the search process used in FLARE different from the dfs used in TOT? It is impressive that the overall cost for computation is low because only 1-3 dfs paths are explored but is this scalable? For example in LogicLM or in F-COT and the use of external solvers would mean that incase of problems having high search depths, these methods would require potentially same number of tokens as now because the number of tokens isn't dependent on search depth. For example in real world SAT problems where search depth is high, one could build a scalable system with LogicLM or F-COT but that would be difficult with FLARE.
> Can you give me a scenario in such cases where it would be prudent to use FLARE instead of F-COT or LogicLM?
>
> 2) Additional Experiments:
> I appreciate the fact that the authors have gone the extra mile to perform experiments on more challenging datasets however here are my concerns.
> Out of 3 datasets FLARE does well on PrOntoQA by a good margin but it does not do well at all on LogicalDeduction by a good margin too. Performance on AR-LSAT is almost close to LogicLM (since it is only a percentage increase)
> One thing where I would conced is the fact that LogicLM uses self refinement as does F-COT while FLARE doesn't. I understand the main contribution fo the paper is to provide an interpretable approach but can you in anyway quantify the effect in performance if FLARE used self refinement?

---

> > ### Author Response · Authors · 2024-11-19
> > **Continuation of Rebuttal for the Reviewer KpGf**
> >
> > ```
> > Additional Experiments: I appreciate the fact that the authors have gone the extra mile to perform experiments on more challenging datasets however here are my concerns. Out of 3 datasets FLARE does well on PrOntoQA by a good margin but it does not do well at all on LogicalDeduction by a good margin too. Performance on AR-LSAT is almost close to LogicLM (since it is only a percentage increase) One thing where I would conced is the fact that LogicLM uses self refinement as does F-COT while FLARE doesn't. I understand the main contribution fo the paper is to provide an interpretable approach but can you in anyway quantify the effect in performance if FLARE used self refinement?
> > ```
> >
> > Per the reviewer's request, we added a simplistic self-refinement on the code generation procedure and tested it on logic-inference benchmarks. We simply prompt the LLM with the previous code in context and ask it to f"Given the error message: {error_message}, please refine the following code: \n{code}". Here, error_message is the error that was obtained during code execution. We complete the self-refinement procedure a maximum of 2 times if the new code is still not executable.
> >
> > |      Dataset     | ChatGPT (gpt-3.5-turbo) |       |          |       |              |
> > |:----------------:|:-----------------------:|:-----:|:--------:|-------|--------------|
> > |   |         Standard        |  CoT  | Logic-LM | FLARE | FLARE_{SR=2} |
> > |     PrOntoQA     |          47.40          | 67.80 |   61.00  | 73.40 |     79.4     |
> > | LogicalDeduction |          40.00          | 42.33 |   65.67  | 58.60 |     64.43    |
> > |      AR-LSAT     |          20.34          | 17.31 |   26.41  | 27.39 |     30.73    |
> >
> > Our results show clear improvements on all of the benchmarks using self-refinement, showing that FLARE benefits from better autoformalizations of a given query. The performance of FLARE can be further improved with better textual representation of the self-refinement task (we used something rather simple) and more refinement iterations, yet already achieves results close to or better than CoT and Logic-LM.

---

> > > ### Author Response · Authors · 2024-11-20
> > > **Continuation of Rebuttal for the Reviewer KpGf**
> > >
> > > ```Search Process: Please correct me if I have misunderstood something. Is the search process used in FLARE different from the dfs used in TOT? It is impressive that the overall cost for computation is low because only 1-3 dfs paths are explored but is this scalable? For example in LogicLM or in F-COT and the use of external solvers would mean that incase of problems having high search depths, these methods would require potentially same number of tokens as now because the number of tokens isn't dependent on search depth. For example in real world SAT problems where search depth is high, one could build a scalable system with LogicLM or F-COT but that would be difficult with FLARE. Can you give me a scenario in such cases where it would be prudent to use FLARE instead of F-COT or LogicLM?```
> > >
> > > The DFS in TOT uses LLM prompting to expand or prune a search path via Value assignment to each state independently or Vote across states. FLARE, on the other hand, simulates prolog execution, which uses DFS by default.
> > >
> > > We do agree that for applications (such as SAT solvers) that require a substantial search depth, the inference cost of the LLM would grow as well. However, knowing that a SAT (or any other external) solver is required (is the tool to use) when the query is asked is already a big contribution towards arriving at a correct answer. Also, correctly formalizing the query into the language the solver supports is a non-trivial task, as seen in Table 1 F-CoT results.
> > >
> > > FLARE is useful for queries that pose ambiguities during the formalization process and require fuzzy, deductive reasoning. Such ambiguities can arise if answering the query requires commonsense, deductive or inductive reasoning.  Fuzzy reasoning with an LLM allows to mitigate such formalization. The system still remains scalable and does not incur additional computational cost for the benchmarks that we tested upoan.
> > >
> > > We respectfully encourage the reviewer to revisit the scores in light of the clarifications, improvements and additional experiments provided in our rebuttal. If any concerns remain, we would greatly appreciate additional feedback or suggestions for improvement.

---

> ### Comment · Reviewer_KpGf · 2024-11-21
>
> After considerable deliberation, I believe that while the increase in inference cost with greater search depth is indeed a drawback, as the authors have pointed out, correctly formalizing a query in the language supported by a solver remains a non-trivial task too.
>
> Following additional experiments, including the integration of self-refinement into FLARE, I find the proposed method to be a noteworthy solver-free, LLM-only approach. The proposed method uniquely combines code generation and execution entirely within the LLM framework, demonstrating significant performance on both arithmetic and complex logical reasoning datasets. The proposed method remains interpretable while achieving results comparable to solver-based approaches like LogicLM.
>
> Minor Suggestions:
> 1) Given ICLR's strict 10-page limit, pls move the reproducibility report to the appendix.
> 2) Pls mention the limitations and future work as it is neccessary.
>
> In light of these considerations, I would recommend the acceptance of this paper.

---

### Official Review · Reviewer_CeSr · 2024-10-30

**Soundness:** 3
**Presentation:** 4
**Contribution:** 3
**Rating:** 6
**Confidence:** 3

**Summary:**

This paper introduces a new method named FLARE for logic-aided reasoning of LLMs using a semi-formal language that is not necessarily executable to allow flexibility. The model is given a problem and proceeds in 3 steps:

1. Generate a natural language plan given few-show examples.
2. Model is prompted to generate a Prolog code given few-shot examples. In this step, the problem is decomposed into set of facts $F$, relations $R$, and a goal $G$.
3. The model is prompted to $\textbf{simulate}$ the execution of the code generated in stage 2 via prompting. This simulation is later used to detect reasoning inconsistencies and measuring faithfulness (to the intermediate reasoning steps) of the model.

There are 2 failure modes in reasoning inconsistencies. First is when a fact or a relation that was not in the Prolog code from step 2 is used in step 3. Second is when a fact or a relation in the code is $\textbf{not}$ used in the reasoning steps, which authors classify as "sub-optimal reasoning."

Authors demonstrate that such prompting with language models such as CmDR and GPT-3.5 results in stronger accuracies on various Q&A and mathematical reasoning benchmarks. The paper has extensive evaluation demonstrating that simulating search and writing code is actually necessary for the performance increase and faithfulness of the generated text/code correlates with the accuracy of the results.

**Strengths:**

- The prompting technique proposed in the paper is very interesting for tasks that are hard to fully formalize yet it can benefit from a semi-formalized approach for logical reasoning.
- The framework seems to reliably improve accuracy on multi-hop QA and MathWord tasks and it is relatively easy to adapt to new problems with just writing few shot examples.
-  The faithfulness metric proposed by the paper seems to correlate well with accuracy which can be used to "weigh" after multiple samples from the model to get more reliable results. This metric also provides a proxy for how reliable the answer is in a very simple way.
- The paper is very well written and easy to read.

**Weaknesses:**

- Although the method seems powerful for QA tasks, its adaptability to harder algorithmic and reasoning tasks seem difficult.
- The authors run experiments with GPT3.5 rather than GPT4o or GPT4o-mini which makes applicability to SoTA models questionable.
- Although the framework increases performance, it does not have the benefits a fully formal system that has guarantees. However, I believe this is okay as they target problems for which it's difficult if not impossible to have guarantees.

**Questions:**

- Could FLARE’s framework extend to other tasks that require complex reasoning? Insights on its limitations in such settings would be helpful.
- The method seems to work more reliably on LLaMa 8b and CmDR rather than on GPT3.5. Is there a reason why?
- The framework seems to fail on sports and date Q&A - some insight into why that happens would be useful.
- Is it true that if the model outputs a Prolog code that is not executable (or wrong syntax), it's not possible do any error detection or faithfulness measure?
- Can you please summarize the 'characteristic' of problems for which the method is applicable and for which kind of problems it is not applicable?

---

> ### Author Response · Authors · 2024-11-19
> **Rebuttal for the Reviewer CeSr**
>
> Rebuttal for  Reviewer CeSr
> We thank the reviewer for their thoughtful and constructive feedback on the paper. Below, we address the raised weaknesses and questions sequentially.
>
>
> **General comments on Weaknesses:**
>
> ```Although the method seems powerful for QA tasks, its adaptability to harder algorithmic and reasoning tasks seem difficult.```
>
> The benchmarks within the study are composed of Math Word Problems, Fuzzy, deductive and relational reasoning challenges like StrategyQA and Clutter. Per the reviewers request, we add a direct comparison with to Logic-LM on the hardest sets of ProntoQA[2], AR-LSAT[3] and Logical Deduction [4]. These datasets are used in the original study of Logic-LM and are considered challenging logic inference benchmarks. We opted not to include PrOntoQA and ProofWriter, as they exhibit considerable overlap with the benchmarks already used in our study (e.g., CLUTTER, AQUA, and StrategyQA). Instead, we prioritised datasets that present unique challenges to demonstrate the efficacy of FLARE better.
>
> | Dataset           | Standard | CoT   | Logic-LM       | FLARE          |
> |-------------------|----------|-------|----------------|----------------|
> | **PrOntoQA**      | 47.40    | 67.80 | 61.00          | **73.40**      |
> | **LogicalDeduction** | 40.00    | 42.33 | **65.67**      | 58.60          |
> | **AR-LSAT**       | 20.34    | 17.31 | 26.41          | **27.39**      |
>
>
> Our findings show that FLARE achieves state-of-the-art results on $2$ out of $3$ logic inference benchmarks with $10\%$ increase over CoT and $~7\%$ increase over Logic-LM. These results highlight FLARE’s ability to handle challenging reasoning tasks compared to existing approaches.
>
>  As logic programming (Prolog) is capable of producing complete algorithms of varying complexities (Anything NP-complete), continuous numbers and nested structures no direct adaptation is needed for more algorithmic tasks.
>
>
> ```The authors run experiments with GPT3.5 rather than GPT4o or GPT4o-mini which makes applicability to SoTA models questionable.```
>
> The main goal of the research was to introduce an interpretable reasoning methodology that is foundational and model-agnostic. We have explored models of varying scales from 8B to 100B+, showing the consistent improvement that FLARE provides to each of them. Given the architectural similarities, the findings from GPT-3.5 are expected to generalize to more advanced models like GPT-4.
>
>
>
> **General comments on Questions**:
>
>
> ```Could FLARE’s framework extend to other tasks that require complex reasoning? Insights on its limitations in such settings would be helpful.```
>
> As mentioned priorly in the comments for weaknesses, FLARE is capable to be extended towards any algorithmic task. Some of the benchmarks already included in the paper require Fuzzy, deductive and relational reasoning. As mentioned, we also added more challenging logic inference benchmarks and showed that FLARE remains efficient in these types of setups as well. In general, we find that FLARE yields the most increase for tasks where formalisation is not straightforward, thus there is a requirement for fuzzy/soft reasoning.
>
> ```The method seems to work more reliably on LLaMa 8b and CmDR rather than on GPT3.5. Is there a reason why?```
>
> GPT-3.5 achieves 4 /9 best results in our benchmarks, demonstrating that using FLARE expands the capabilities of the model. Furthermore, our method consistently outperforms CoT and F-CoT in 5/9 benchmarks. Notably, it surpasses F-CoT across all benchmarks.
>
> ```The framework seems to fail on sports and date Q&A - some insight into why that happens would be useful.```
>
> The two benchmarks where CoT somewhat outperforms our method—GSM8k and Sports—are tasks that primarily require straightforward one-hop reasoning or retrieval from a commonsense knowledge base. These tasks do not significantly benefit from the autoformalization and search mechanisms central to our method, which may explain the observed differences.
>
> ```Is it true that if the model outputs a Prolog code that is not executable (or wrong syntax), it's not possible do any error detection or faithfulness measure?```
>
> It is possible to detect underutilizations or hallucinations within the search w.r.t. the written code regardless if it is executable or not. It is also possible to detect if the code is correct and try to refine it if there is an error, however such approaches are rather computationally costly and are not a part of the original study. It is however not possible to measure the faithfulness of the sample without executable code, as the trace generated from the original code, which we compare to the LLM simulation, can only be generated in by running the code.
>
> We hope that the clarifications and the questions addressed in our rebuttal have resolved any concerns, allowing the reviewer to consider increasing the support for the paper. If there are any remaining issues, we welcome further feedback for improvement.

---

> > ### Author Response · Authors · 2024-11-20
> >
> > Thank you for your valuable feedback on our submission. We’ve addressed your comments and provided detailed clarifications in our rebuttal. We would greatly appreciate it if you could take a moment to review our responses and share any further thoughts or follow-up questions.
> >
> > We hope that the clarifications provided and the questions addressed in our rebuttal have resolved any concerns, allowing the reviewer to consider increasing the support for the paper. If there are any remaining issues, we welcome further feedback or suggestions for improvement.

---

> > > ### Comment · Reviewer_CeSr · 2024-11-26
> > >
> > > ```The main goal of the research was to introduce an interpretable reasoning methodology that is foundational and model-agnostic. We have explored models of varying scales from 8B to 100B+, showing the consistent improvement that FLARE provides to each of them. Given the architectural similarities, the findings from GPT-3.5 are expected to generalize to more advanced models like GPT-4.```
> > >
> > > I disagree with this statement - if the method is helpful when the model is not capable of some things yet, doesn't necessarily mean that it will be useful when the model is more advanced. Can you reproduce the experiments with GPT4o for comparison?

---

> > > > ### Author Response · Authors · 2024-11-26
> > > > **Continuation of Rebuttal for the Reviewer CeSr**
> > > >
> > > > Per the reviewers request, we add a direct comparison using GPT-4o and GPT3.5 with to Logic-LM and CoT on the hardest sets of ProntoQA[2], AR-LSAT[3] and Logical Deduction [4]. These datasets are used in the original study of Logic-LM and are considered challenging logic inference benchmarks. We opted not to include FOLIO and ProofWriter, as they exhibit considerable overlap with the benchmarks already used in our study (e.g., CLUTTER, AQUA, and StrategyQA). Instead, we prioritised datasets that present unique challenges to demonstrate the efficacy of FLARE better.
> > > >
> > > > The reviewer `KpGf` pointed out, "LogicLM uses self-refinement, as does F-COT, while FLARE doesn't." Thus, we also added self-refinement to the code generation procedure and tested it on logic-inference benchmarks (marked as SR in the results). Please see the complete thread with the reviewer `KpGf` for the complete context.
> > > >
> > > > |                  | **_GPT-4 (gpt-4o)_** |       |          |       |              |
> > > > |:----------------:|:--------------------:|:-----:|:--------:|-------|--------------|
> > > > |   **_Dataset_**  |       Standard       |  CoT  | Logic-LM | FLARE | FLARE_{SR=2} |
> > > > |     PrOntoQA     |         77.40        | 98.79 |   83.20  | *98.87* | **99.24**    |
> > > > | LogicalDeduction |         71.33        | 75.25 |   87.63  | *88.0*  | **90.33**    |
> > > > |      AR-LSAT     |         33.33        | 35.06 |   43.04  | *39.82* | **45.02**    |
> > > >
> > > >
> > > > |      Dataset     | ChatGPT (gpt-3.5-turbo) |       |          |       |              |
> > > > |:----------------:|:-----------------------:|:-----:|:--------:|-------|--------------|
> > > > |   |         Standard        |  CoT  | Logic-LM | FLARE | FLARE_{SR=2} |
> > > > |     PrOntoQA     |          47.40          | 67.80 |   61.00  | 73.40 |     **79.4**     |
> > > > | LogicalDeduction |          40.00          | 42.33 |   65.67  | 58.60 |     **64.43**    |
> > > > |      AR-LSAT     |          20.34          | 17.31 |   26.41  | 27.39 |     **30.73**    |
> > > >
> > > > Our results show clear improvements on all of the benchmarks with the GPT-4o use and are even better with added self-refinement, showing that FLARE benefits from better autoformalizations of a given query. The performance of FLARE can be further improved with better textual representation of the self-refinement task (we used something rather simple as pointed out in the thread with `KpGf`) and more refinement iterations, yet already achieves results better than CoT and Logic-LM.
> > > >
> > > > We hope that the clarifications provided and the questions addressed in our rebuttal have resolved any concerns, allowing the reviewer to consider increasing the support for the paper. If there are any remaining issues, we welcome further feedback or suggestions for improvement.

---

> > > > > ### Author Response · Authors · 2024-11-27
> > > > >
> > > > > Thank you for your valuable feedback on our submission. We’ve addressed your comments and provided detailed clarifications in our rebuttal. We would greatly appreciate it if you could take a moment to review our responses and share any further thoughts or follow-up questions.
> > > > >
> > > > > We hope that the clarifications provided and the questions addressed in our rebuttal have resolved any concerns, allowing the reviewer to consider increasing the support for the paper. If there are any remaining issues, we welcome further feedback or suggestions for improvement.

---

> > > > > > ### Author Response · Authors · 2024-11-29
> > > > > >
> > > > > > Thank you for your valuable feedback on our submission. We’ve addressed your comments and provided detailed clarifications in our rebuttal. We would greatly appreciate it if you could take a moment to review our responses and share any further thoughts or follow-up questions.
> > > > > >
> > > > > > We hope that the clarifications provided and the questions addressed in our rebuttal have resolved any concerns, allowing the reviewer to consider increasing the support for the paper. If there are any remaining issues, we welcome further feedback or suggestions for improvement.

---

> > > > > > > ### Author Response · Authors · 2024-12-01
> > > > > > >
> > > > > > > Thank you for your valuable feedback on our submission. We’ve addressed your comments and provided detailed clarifications in our rebuttal. We hope that the clarifications provided and the questions addressed in our rebuttal have resolved any concerns, allowing the reviewer to consider increasing the support for the paper.

---

### Official Review · Reviewer_VyMj · 2024-10-31

**Soundness:** 4
**Presentation:** 2
**Contribution:** 3
**Rating:** 6
**Confidence:** 5

**Summary:**

This paper presents a reasoning framework capable of answering fuzzy and multi-hop questions while ensuring transparency in its reasoning steps. The framework achieves this through three key phases: task decomposition, soft formalization, and simulated search. In the first phase, the LLM is prompted to analyze the question and generate a structured plan for subsequent formalization. Following this, the LLM translates the question into ProLog programs according to the provided plan, called "soft formalization." Finally, the framework searches for answers by simulating a depth-first search over the problem space, essentially emulating Prolog execution within the LLM. This structured pipeline allows the framework to detect inconsistencies in reasoning by examining the facts and relationships used during inference. Additionally, it assesses the faithfulness of the reasoning process by comparing the actual execution paths to simulated paths generated by the LLM. Their methods achieved SOTA performance on 7 out of 9 common reasoning benchmarks.

**Strengths:**

1. The authors address limitations in current methods, which either struggle to maintain faithful, consistent reasoning steps or fail to handle fuzzy reasoning for difficult-to-formalize problems. They tackle these issues with a soft-formalization approach and a simulated search mechanism.
2. Unlike F-CoT, their approach doesn’t require programming expertise or additional fine-tuning for the LLMs. The LLMs aren’t required to generate perfectly accurate code but instead simulate Prolog execution internally, bypassing the need for an external interpreter.
3. The use of Prolog encourages the LLM to explicitly outline facts and relations in its reasoning, enhancing control and enabling verification of correct fact and relation usage, which helps reduce hallucinations.
4. Prolog’s DFS feature ensures thorough exploration of the problem space, increasing the likelihood of finding the correct answer.
5. The experiments are well-designed and thorough, demonstrating the method’s effectiveness with results that surpass or match those of CoT and F-CoT. The ablation study results for simulated search align well with expectations.
6. The correlation between faithfulness and reasoning performance is effectively highlighted through experimental results, supporting the model’s interpretability and reliability.

**Weaknesses:**

1. The paper would benefit from more detailed examples to illustrate how the simulated search operates and how it benefits intermediate reasoning steps.
2. There is no theoretical analysis of search complexit. If complexity analysis isn’t feasible for certain fuzzy problems, a time cost comparison between FLARE and baseline methods (e.g., CoT and F-CoT) would be a useful alternative.

**Questions:**

1. Considering that DFS can become computationally intensive as search depth increases, and that appending search traces to prompts may risk exceeding the LLMs' context length, how do you manage the generated Prolog program to avoid an unacceptable search space?
2. Would it be possible to set a faithfulness threshold and continue the simulated search only until this threshold is met (i.e., early stopping)? Testing this approach could make the way to faithfulness calculation more convincing.

---

> ### Author Response · Authors · 2024-11-19
> **Rebuttal for the Reviewer VyMj**
>
> We thank the reviewer for their thought-provoking feedback on the paper and proceed to sequentially address the raised Weaknesses and Questions:
>
> General comments on Weaknesses:
>
> ```The paper would benefit from more detailed examples to illustrate how the simulated search operates and how it benefits intermediate reasoning steps.```
>
> We have updated Section 3 and 5 with more explanations and demonstrations to clarify the simulated search process and the benefits that it offers. We also further refine Figure 1, which showcases an example of simulated search and the hallucination and underutilisation detection process. We also added a raw complete example in the Appendix for observation. We also include all the prompts and solved samples for each dataset within our anonymised repo: https://anonymous.4open.science/r/FLARE-D88B
>
> ```There is no theoretical analysis of search complexity. If complexity analysis isn’t feasible for certain fuzzy problems, a time cost comparison between FLARE and baseline methods (e.g., CoT and F-CoT) would be a useful alternative.```
>
> The speed of the external solvers used for baseline methods (F-CoT) is highly dependent and sensitive towards the executing system and choice of the setup that runs it (F-CoT can be run with different versions of Python, PDDL etc). This makes a direct comparison with LLM inference impossible. However, per the reviewer's request, we compare the number of tokens and the inference speed for generating CoT vs F-CoT (without the external solver added time) vs FLARE.
>
> | model          |   ('plan', 'mean') |   ('plan', 'std') |   ('code', 'mean') |   ('code', 'std') |   ('trace', 'mean') |   ('trace', 'std') |   ('trace_percentage', 'mean') |   ('trace_percentage', 'std') |
> |:---------------|-------------------:|------------------:|-------------------:|------------------:|--------------------:|-------------------:|-------------------------------:|------------------------------:|
> | Llama-3.1-8B   |            552.544 |           146.769 |            241.274 |           128.231 |             336.255 |            281.753 |                        26.9211 |                       11.3238 |
> | command-r      |            511.435 |           160.666 |            180.228 |           132.454 |             266.82  |            268.343 |                        23.0468 |                       15.1715 |
> | command-r-plus |            577.032 |           184.77  |            189.203 |           151.211 |             223.105 |            209.69  |                        19.7404 |                       12.5113 |
> | gpt-3.5-turbo  |            557.774 |           161.733 |            177.533 |           128.255 |             164.41  |            184.529 |                        16.3258 |                       10.686  |
>
>
> Our results show that the total length of the trace is on average composed of ~250 tokens which corresponds to an approximately 20% increase compared to CoT and F-CoT. Despite this marginal increase, the impact on overall inference speed is negligible. In particular, the additional tokens are not a reason for a computational overhead as the computational cost of adding a small number of tokens (such as the additional ~250 tokens in FLARE traces) is minimal compared to the overall cost of processing the base sequence. This is specifically true when the sequence lengths remain well below the model's maximum context window.
>
>
>
> General comments on Questions:
>
> ```Considering that DFS can become computationally intensive as search depth increases, and that appending search traces to prompts may risk exceeding the LLMs' context length, how do you manage the generated Prolog program to avoid an unacceptable search space?```
>
> As shown above, the overall length of the average plan, code and search remains below 1000 tokens, and with added 4-8 in context samples, would still remain well under the context length limit (128K for most models, 16K for GPT3.5).
>
> We also have provided the overall statistics for the completed search in Table 4 within the paper. Our results show that while the models are capable of performing 11 multi-hop steps within a single DFS branch/path, usually only 1-3 paths are explored for a total of 250 tokens on average. We see a similar distribution from the traces generated from actual code execution. We postulate that most of the queries do not require the depth of the search to be high and also show that models use fuzzy reasoning to skip failing paths and reasoning lines (Tables 4, 5,6), thus truncating the need for exhaustive search.

---

> > ### Author Response · Authors · 2024-11-19
> > **Continuation of the Rebuttal for the Reviewer VyMj**
> >
> > ```Would it be possible to set a faithfulness threshold and continue the simulated search only until this threshold is met (i.e., early stopping)? Testing this approach could make the way to faithfulness calculation more convincing.```
> >
> > This is possible in cases where we obtain runnable code, which is not required in our framework. We want to clarify that faithfulness measurement is done after inference on a sample/benchmark is completed, because perpetually checking if the faithfulness score has reached a threshold is computationally costly. While thresholding, self-refinement and other techniques can be applied on top of FLARE it is not a part of the original motivation of the study as the goal was to create an interpretable method that allows for planning, fuzzy reasoning, and traversing the problem space with backtracking, exact task decomposition, and measuring faithfulness.
> >
> > We hope that the clarifications provided and the questions addressed in our rebuttal have resolved any concerns, allowing the reviewer to consider increasing the support for the paper. If there are any remaining issues, we welcome further feedback or suggestions for improvement.

---

> > > ### Author Response · Authors · 2024-11-20
> > >
> > > Thank you for your valuable feedback on our submission. We’ve addressed your comments and provided detailed clarifications in our rebuttal. We would greatly appreciate it if you could take a moment to review our responses and share any further thoughts or follow-up questions.
> > >
> > > We hope that the clarifications provided and the questions addressed in our rebuttal have resolved any concerns, allowing the reviewer to consider increasing the support for the paper. If there are any remaining issues, we welcome further feedback or suggestions for improvement.

---

> > > > ### Author Response · Authors · 2024-11-28
> > > >
> > > > Thank you for your valuable feedback on our submission. We’ve addressed your comments and provided detailed clarifications in our rebuttal. We would greatly appreciate it if you could take a moment to review our responses and share any further thoughts or follow-up questions.
> > > >
> > > > Per the reviewers request, we add a direct comparison using GPT-4o and GPT3.5 with to Logic-LM and CoT on the hardest sets of ProntoQA[2], AR-LSAT[3] and Logical Deduction [4]. These datasets are used in the original study of Logic-LM and are considered challenging logic inference benchmarks. We opted not to include FOLIO and ProofWriter, as they exhibit considerable overlap with the benchmarks already used in our study (e.g., CLUTTER, AQUA, and StrategyQA). Instead, we prioritised datasets that present unique challenges to demonstrate the efficacy of FLARE better.
> > > >
> > > > The reviewer `KpGf` pointed out, "LogicLM uses self-refinement, as does F-COT, while FLARE doesn't." Thus, we also added self-refinement to the code generation procedure and tested it on logic-inference benchmarks (marked as SR in the results). Please see the complete thread with the reviewer `KpGf` for the complete context.
> > > >
> > > > |                  | **_GPT-4 (gpt-4o)_** |       |          |       |              |
> > > > |:----------------:|:--------------------:|:-----:|:--------:|-------|--------------|
> > > > |   **_Dataset_**  |       Standard       |  CoT  | Logic-LM | FLARE | FLARE_{SR=2} |
> > > > |     PrOntoQA     |         77.40        | 98.79 |   83.20  | *98.87* | **99.24**    |
> > > > | LogicalDeduction |         71.33        | 75.25 |   87.63  | *88.0*  | **90.33**    |
> > > > |      AR-LSAT     |         33.33        | 35.06 |   43.04  | *39.82* | **45.02**    |
> > > >
> > > >
> > > > |      Dataset     | ChatGPT (gpt-3.5-turbo) |       |          |       |              |
> > > > |:----------------:|:-----------------------:|:-----:|:--------:|-------|--------------|
> > > > |   |         Standard        |  CoT  | Logic-LM | FLARE | FLARE_{SR=2} |
> > > > |     PrOntoQA     |          47.40          | 67.80 |   61.00  | 73.40 |     **79.4**     |
> > > > | LogicalDeduction |          40.00          | 42.33 |   65.67  | 58.60 |     **64.43**    |
> > > > |      AR-LSAT     |          20.34          | 17.31 |   26.41  | 27.39 |     **30.73**    |
> > > >
> > > > Our results show clear improvements on all of the benchmarks with the GPT-4o use and are even better with added self-refinement, showing that FLARE benefits from better autoformalizations of a given query. The performance of FLARE can be further improved with better textual representation of the self-refinement task (we used something rather simple as pointed out in the thread with `KpGf`) and more refinement iterations, yet already achieves results better than CoT and Logic-LM.
> > > >
> > > > We hope that the clarifications provided and the questions addressed in our rebuttal have resolved any concerns, allowing the reviewer to consider increasing the support for the paper. If there are any remaining issues, we welcome further feedback or suggestions for improvement.

---

> > > > > ### Author Response · Authors · 2024-11-29
> > > > >
> > > > > Thank you for your valuable feedback on our submission. We’ve addressed your comments and provided detailed clarifications in our rebuttal. We would greatly appreciate it if you could take a moment to review our responses and share any further thoughts or follow-up questions.
> > > > >
> > > > > We hope that the clarifications provided and the questions addressed in our rebuttal have resolved any concerns, allowing the reviewer to consider increasing the support for the paper. If there are any remaining issues, we welcome further feedback or suggestions for improvement.

---

> > > > > > ### Author Response · Authors · 2024-12-01
> > > > > >
> > > > > > Thank you for your valuable feedback on our submission. We’ve addressed your comments and provided detailed clarifications in our rebuttal. We hope that the clarifications provided and the questions addressed in our rebuttal have resolved any concerns, allowing the reviewer to consider increasing the support for the paper.

---

### Official Review · Reviewer_ycsU · 2024-11-02

**Soundness:** 2
**Presentation:** 2
**Contribution:** 2
**Rating:** 3
**Confidence:** 4

**Summary:**

This paper introduces Faithful Logic-Aided Reasoning and Exploration (FLARE) to enhance the interpretability and faithfulness for LLM-based reasoning. FLARE uses task decomposition and logic programming to guide LLMs. It combines planning with logical reasoning, where the query is formulated into structured facts and predicates, and then explored through an exhaustive multi-hop search within a defined problem space. At the end, there are some reasoning inconsistencies detection and faithfulness measurement.

**Strengths:**

Improving LLM-based reasoning with neuro-symbolic integration is a good research problem. Empirical results are given.

**Weaknesses:**

The writing could benefit from adjustments to improve clarity, strengthen notation, and correct typos. The advantages of using LLMs to simulate Prolog execution are ambiguous. Some key design and implementation details are missing. Code and data are not provided for reproducibility.

**Questions:**

1) It would be helpful to include the notations for equations (Eq. (1)-(4)) and explanations for figures (e.g., Figure 1) to enhance readability and logical flow. Additionally, there are a few specific points that may benefit from clarification. For example, $T$ as a single token in query $Q$ is misused in Eq. (1)-(4). In Eq. (4), there might be a redundant comma or a missing variable.

2) In Section 3.1, the Simulating Search section might benefit from additional details. Specifically, elaborating on how a problem space traversal trace is formed would be valuable. Including a detailed description, pseudocode, or examples could provide further clarity. It would also be helpful to include a detailed explanation of the backtracking mechanism.

3) The complete process could be further clarified by explaining the steps following inconsistency detection and faithfulness measurement. For example, it would be valuable to know if a best-of-n selection, weighted voting or other strategies are used to decide the final answer. A pseudocode of the entire framework could also enhance understanding here. In addition, it's better to include some full examples for inconsistency detection and faithfulness measurement.

4) The framework may benefit from adaptive capabilities. Currently, the plan generation is done in one pass without a feedback loop. Introducing a mechanism to refine the generated code if errors are detected during subsequent simulation could also improve adaptability.

5) For simulating Prolog execution, the choice to use LLMs over an external solver might benefit from further clarification. Understanding the reasoning behind using an LLM-based simulation instead of an external solver, particularly in terms of handling the self-bias of LLMs, would be insightful.

6) Since Prolog is tailored for rule-based logical reasoning, the specific adaptations for solving math word problems need to be clarified. Further implementation details and example outputs could illustrate the achieved improvements and highlight the framework's effectiveness.

7) It would be valuable to expand the evaluation to include other general methods, such as Tree of Thought, Program of Thought, and Self-Consistency, alongside comparisons to similar methods [1-3]. Additionally, for Llama-3.1-8B, CmDR, and CMDR+, the F-CoT results are zero. Could the authors provide more context on this outcome?

8) Code and data are not provided for reproducibility.

Reference:

[1] Pan, Liangming, et al. "Logic-LM: Empowering Large Language Models with Symbolic Solvers for Faithful Logical Reasoning." The 2023 Conference on Empirical Methods in Natural Language Processing.

[2] Yang, Sen, et al. "Neuro-symbolic integration brings causal and reliable reasoning proofs." arXiv preprint arXiv:2311.09802 (2023).

[3] Xu, Fangzhi, et al. "Symbol-LLM: Towards foundational symbol-centric interface for large language models." arXiv preprint arXiv:2311.09278 (2023).

---

> ### Author Response · Authors · 2024-11-18
> **Rebuttal for  Reviewer ycsU**
>
> We thank the reviewer for their feedback and questions and proceed to sequentially address the raised Weaknesses and Questions:
>
> **General comments on Weaknesses**:
>
> `The advantages of using LLMs to simulate Prolog execution are ambiguous. `
>
> Our results show that the models simulating logic programming execution yield an average of increase of 23% over CoT and ~60% on F-CoT and yield SOTA results across 7 of the 9 QA tasks that we tested on. We have further upon the reviewers request added 3 logic inference benchmarks ProntoQA, AR-LSAT, LogicalDeduction [1,2,3] on which the model shows an average 16% increase over CoT and 8% increase over Logic-LM on 2 out of 3 benchmarks. We show that the increase in results are due to the presence of those simulated code execution by directly comparing performance with plan-only and code approaches (Section 5.2).
>
> The method also allows to explicitly measure the faithfulness of the final answer w.r.t. the performed multi-hop search. Furthermore, FLARE allows direct detection of hallucinations and underutilised knowledge during the search process, which is a unique feature of a reasoning method allowed through code execution simulation. We are able to confirm this way that the optimal searches exhibit, on average, a $18.1\%$ increase in unique emergent facts, a $8.6\%$ higher overlap between code-defined and execution-trace relations, and a $3.6\%$ reduction in unused code relations.
> We show that this behaviour is very specific
>
>
> `Some key design and implementation details are missing. Code and data are not provided for reproducibility.`
>
> We have added all of the implementation details, all of the used prompts and results in the submitted updated paper, supplementary materials and the code. All of these can also be found in our anonymously shared code repository: https://anonymous.4open.science/r/FLARE-D88B/README.md
>
> **Answer to Questions**:
>
> ```It would be helpful to include the notations for equations (Eq. (1)-(4)) and explanations for figures (e.g., Figure 1) to enhance readability and logical flow. Additionally, there are a few specific points that may benefit from clarification. For example, T as a single token in query Q is misused in Eq. (1)-(4). In Eq. (4), there might be a redundant comma or a missing variable.```
>
> We have updated the readability of the overall paper, including the notation, equations, their definitions, demonstrations and the overall flow of the paper. (See updated manuscript)
>
> ```In Section 3.1, the Simulating Search section might benefit from additional details. Specifically, elaborating on how a problem space traversal trace is formed would be valuable. Including a detailed description, pseudocode, or examples could provide further clarity. It would also be helpful to include a detailed explanation of the backtracking mechanism.```
>
> We have added a detailed explanation of the whole search process in the section and refined the writing with additional details. We further improved Figure 1 which includes a direct demonstration of the whole search process with backtracking. We have also added a complete example of FLARE in the Appendix of the paper.
>
> ```The complete process could be further clarified by explaining the steps following inconsistency detection and faithfulness measurement. For example, it would be valuable to know if a best-of-n selection, weighted voting or other strategies are used to decide the final answer. A pseudocode of the entire framework could also enhance understanding here. In addition, it's better to include some full examples for inconsistency detection and faithfulness measurement.```
>
> We want to clarify that faithfulness measurement and hallucination detection are done after inference on a benchmark is completed. This means that they do not impact the final answer-generation process. In FLARE, the final answer is generated by directly prompting the LLM as explained in Section 3.1. We further refined that sections 3.2 and 3.3 for a more clear description of the faithfulness measurement and hallucination detection. We also improved figure 1 to explicitly show how the matching for hallucinated and underutilised facts and relations are done for a given example.

---

> ### Author Response · Authors · 2024-11-18
> **Continuation for the Rebuttal for Reviewer ycsU**
>
> ```The framework may benefit from adaptive capabilities. Currently, the plan generation is done in one pass without a feedback loop. Introducing a mechanism to refine the generated code if errors are detected during subsequent simulation could also improve adaptability.```
>
> While self-refinement and other techniques can be applied on top of FLARE it is not a part of the original motivation of the study as the goal was to create an interpretable method that allows for planning, fuzzy reasoning, and traversing the problem space with backtracking, exact task decomposition, and measuring faithfulness.
>
> It must also be noted that the suggested self-refinement loops are highly computationally costly, but must not necessarily improve the over performance of the method, as FLARE does not rely on the correctness of the generated code.
>
>
> ```For simulating Prolog execution, the choice to use LLMs over an external solver might benefit from further clarification. Understanding the reasoning behind using an LLM-based simulation instead of an external solver, particularly in terms of handling the self-bias of LLMs, would be insightful.```
>
> Directly using external solvers comes with a variety of limitations.
>
> 1. Not all natural language queries can be safely and correctly translated into a formal language (Autoformalization [1]), because queries leave off crucial commonsense, domain or deductive information, thus not allowing either to produce an executable code or code that contains expressive enough facts, relation or strategies for reasoning. This is discussed in Sections 5.1 and 5.2.
>
> 2. External solvers are also incapable of fuzzy reasoning and reasoning outside of the scope defined within the program, while LLM simulated code executions do not have such a limitation.
> This means that queries involving contextual ambiguity and requiring deductive or analytical reasoning not bounded by a predefined scope benefit from the use of LLMs for simulated code execution.
>
> ```Since Prolog is tailored for rule-based logical reasoning, the specific adaptations for solving math word problems need to be clarified. Further implementation details and example outputs could illustrate the achieved improvements and highlight the framework's effectiveness.```
>
> Prolog directly supports mathematical operations and is capable of outputting continuous numbers in a program. In this sense no specific adaptation is needed for solving math world problems. We include all the prompts and solved samples for each dataset within our  anonymized repo: https://anonymous.4open.science/r/FLARE-D88B
>
> We also refined Section 3 for further clarifications.
>
>
> ```It would be valuable to expand the evaluation to include other general methods, such as Tree of Thought, Program of Thought, and Self-Consistency, alongside comparisons to similar methods [1-3]. Additionally, for Llama-3.1-8B, CmDR, and CMDR+, the F-CoT results are zero. Could the authors provide more context on this outcome?```
>
>
> Per the reviewer's request, we add a direct comparison with Logic-LM on the hardest sets of ProntoQA[2], AR-LSAT[3] and Logical Deduction [4]. These datasets are used in the original study of Logic-LM and are considered challenging logic inference benchmarks. We do not use FOLIO and ProofWriter as they are rather similar to the currently existing benchmarks (Clutter and StrategyQA) in our study.
>
> | Dataset           | Standard | CoT   | Logic-LM       | FLARE          |
> |-------------------|----------|-------|----------------|----------------|
> | **PrOntoQA**      | 47.40    | 67.80 | 61.00          | **73.40**      |
> | **LogicalDeduction** | 40.00    | 42.33 | **65.67**      | 58.60          |
> | **AR-LSAT**       | 20.34    | 17.31 | 26.41          | **27.39**      |
>
> **Table**: Comparison of Direct Prompting, CoT, Logic-LM, and FLARE.
>
> Our findings show that FLARE achieves state-of-the-art results on $2$ out of $3$ logic inference benchmarks with $10\%$ increase over CoT and $~7\%$ increase over Logic-LM.
>
> As the Tree of Thoughts and Self-Consistency do not offer ways for measuring faithfulness and do not necessarily produce interpretable answers that follow a given search logic over a defined problem space (Facts, relation, search goal) while being extremely computationally costly, comparing to them would be outside of the scope of the current research and would not add any new revelations.
>
>
> ```Code and data are not provided for reproducibility.```
>
> We provide the complete code and data for the reproduction on the experiments in an anonymised repository: https://anonymous.4open.science/r/FLARE-D88B
>
>
> We respectfully request the reviewer to kindly reassess the scores and consider increasing their support for the paper unless they have remaining concerns with our work or the rebuttal, in which case we welcome any additional feedback or clarifications.

---

> > ### Author Response · Authors · 2024-11-19
> > **Contiunuation of the Rebuttal**
> >
> > ```The F-CoT results are zero. Could the authors provide more context on this outcome?```
> >
> > As clarified in the paper, F-CoT requires executable code with a specific structure to be able to produce a final prediction. However, models not explicitly tuned for such constrained coding tasks fail to produce an output that is parseable, thus resulting in degenerate answers for a big portion of solvers. We tested using the original F-CoT prompts and codebase.

---

> > ### Author Response · Authors · 2024-11-19
> > **Continuation for the Rebuttal for Reviewer ycsU**
> >
> > ```The framework may benefit from adaptive capabilities.```
> >
> > Per the reviewer's request, we also added a simplistic self-refinement on the code generation procedure and tested it on logic-inference benchmarks. We simply prompt the LLM with the previous code in context and ask it to f"Given the error message: {error_message}, please refine the following code: \n{code}". Here, error_message is the error that was obtained during code execution. We complete the self-refinement procedure a maximum of 2 times if the new code is still not executable.
> >
> > |      Dataset     | ChatGPT (gpt-3.5-turbo) |       |          |       |              |
> > |:----------------:|:-----------------------:|:-----:|:--------:|-------|--------------|
> > |   |         Standard        |  CoT  | Logic-LM | FLARE | FLARE_{SR=2} |
> > |     PrOntoQA     |          47.40          | 67.80 |   61.00  | 73.40 |     79.4     |
> > | LogicalDeduction |          40.00          | 42.33 |   65.67  | 58.60 |     64.43    |
> > |      AR-LSAT     |          20.34          | 17.31 |   26.41  | 27.39 |     30.73    |
> >
> > Our results show clear improvements on all of the benchmarks using self-refinement, showing that FLARE benefits from better autoformalizations of a given query. The performance of FLARE can be further improved with better textual representation of the self-refinement task (we used something rather simple) and more refinement iterations, yet already achieves results close to or better than CoT and Logic-LM.
> >
> > We respectfully request the reviewer to kindly reassess the scores and consider increasing their support for the paper unless they have remaining concerns with our work or the rebuttal, in which case we welcome any additional feedback or clarifications.

---

> > > ### Author Response · Authors · 2024-11-20
> > >
> > > Thank you for your valuable feedback on our submission. We’ve addressed your comments and provided detailed clarifications in our rebuttal. We would greatly appreciate it if you could take a moment to review our responses and share any further thoughts or follow-up questions.
> > >
> > > We respectfully request the reviewer to kindly reassess the scores and consider increasing their support for the paper unless they have remaining concerns with our work or the rebuttal, in which case we welcome any additional feedback or clarifications.

---

> > > > ### Author Response · Authors · 2024-11-22
> > > >
> > > > Thank you for your valuable feedback on our submission. We’ve addressed your comments and provided detailed clarifications in our rebuttal. We would greatly appreciate it if you could take a moment to review our responses and share any further thoughts or follow-up questions.
> > > >
> > > > We respectfully request the reviewer to kindly reassess the scores and consider increasing their support for the paper unless they have remaining concerns with our work or the rebuttal, in which case we welcome any additional feedback or clarifications.

---

> > > ### Comment · Reviewer_ycsU · 2024-11-23
> > > **Official Comment by Reviewer ycsU**
> > >
> > > I appreciate the detailed rebuttals by the authors. However, I still have the following concerns.
> > >
> > > ## Novelty
> > > The proposed framework is purely based on prompting with a straightforward design. The approach of performing reasoning without relying on external tools has been explored in related work [1,2,3]. Moreover, numerous LLM-based agentic frameworks for reasoning already exist. Consequently, the novelty of this work appears to be limited.
> > >
> > > ## Clarity
> > > As noted in the initial review, the methodology section remains unclear. The simulating search described in Section 3.1 only covers the generation of a single trace, leaving it unclear how the authors handle path searching and backtracking—arguably the most critical aspects. No detailed pseudo-code is provided to clarify these processes. In contrast, other search-based algorithms, such as Tree of Thought (ToT) and RAP [4], typically include clear pseudo-code in their papers to aid understanding.
> > >
> > > ## Performance discrepancies
> > > The reported results for Llama-3.1-8B CoT are significantly lower than those presented in the official Llama paper [5] and other studies [6]. For example, its performance on GSM8k is reported as 74.5 in [6] but only 59.2 in this paper; on SVAMP, it is 89.2 in [6] but 58.6 here; and on StrategyQA, it is 68.4 in [6] but 2.9 in this paper. A similar issue is observed with GPT-3.5 CoT. These discrepancies raise concerns about the validity of the evaluation conducted by the authors, leaving the claimed superiority of the proposed method over CoT unclear.
> > >
> > > ## Comparison to related work
> > > As noted in the initial review, the comparison to related work remains very limited. The proposed method simulates program execution by searching in the problem space, yet it does not include comparisons with classic search algorithms such as Tree of Thought (ToT) and Self-Consistency (SC). Comparing a search-based approach solely with one-pass CoT is insufficient and unfair.
> > >
> > > Additionally, the primary baseline used for comparison is F-CoT [7], but there is a scope mismatch. F-CoT primarily targets logical reasoning tasks (e.g., ProntoQA, ProofWriter, FOLIO, LogicalDeduction, AR-LSAT), whereas the benchmarks in this paper focus on math problems and multi-hop QA. It is unclear whether the authors have carefully adapted F-CoT to these benchmarks, raising concerns about the validity of the comparisons.
> > >
> > > ## Significance
> > > The paper fails to justify the pure prompting-based design, as it essentially renders the approach into yet another fancy CoT or ToT method that could hallucinate during its reasoning. Additionally, as noted in the initial review, language models (LMs) are generally more effective at handling natural language than symbolic representations. Integrating symbolic representation into the reasoning process of LMs solely through prompting is likely to degrade their performance, which is shown in [8].
> > >
> > > Even from a pure performance perspective, the baseline results are limited and questionable, making the significance of this work appear to be minor.
> > >
> > > ## Reference:
> > >
> > > [1] Zhu, Zhaocheng, et al. "Large language models can learn rules." arXiv preprint arXiv:2310.07064 (2023).
> > >
> > > [2] Feng, Jiazhan, et al. "Language models can be logical solvers." arXiv preprint arXiv:2311.06158 (2023).
> > >
> > > [3] Li, Qingchuan, et al. "Leveraging LLMs for Hypothetical Deduction in Logical Inference: A Neuro-Symbolic Approach." arXiv preprint arXiv:2410.21779 (2024).
> > >
> > > [4] Hao, Shibo, et al. "Reasoning with language model is planning with world model." arXiv preprint arXiv:2305.14992 (2023).
> > >
> > > [5] Dubey, Abhimanyu, et al. "The llama 3 herd of models." arXiv preprint arXiv:2407.21783 (2024).
> > >
> > > [6] Qi, Zhenting, et al. "Mutual reasoning makes smaller llms stronger problem-solvers." arXiv preprint arXiv:2408.06195 (2024).
> > >
> > > [7] Xu, Jundong, et al. "Faithful Logical Reasoning via Symbolic Chain-of-Thought." arXiv preprint arXiv:2405.18357 (2024).
> > >
> > > [8] Tam, Zhi Rui, et al. "Let me speak freely? a study on the impact of format restrictions on performance of large language models." arXiv preprint arXiv:2408.02442 (2024).

---

> ### Author Response · Authors · 2024-11-24
> **Continuation for the Rebuttal for Reviewer ycsU**
>
> We would like to address all of the points made by the reviewer and point out several conceptual misunderstandings and missed/incorrect benchmark comparisons in the paper.
>
> ```Novelty: The proposed framework is purely based on prompting with a straightforward design. The approach of performing reasoning without relying on external tools has been explored in related work [1,2,3].```
>
> The goal was to create an interpretable method that allows for planning, fuzzy reasoning, and traversing the problem space with backtracking, exact task decomposition, and measuring faithfulness.
>
> None of the pointed-out studies explore logic program simulation with an LLM and address faithfulness measurement, hallucination and suboptimal reasoning detection. Also, all of these papers use vastly different methodologies and pursue different goals.
>
>
> Let’s go through each of them:
>  1. [1] “Large Language Models can Learn Rules” - This paper proposes a straightforward LLM prompting inductive-to-deductive reasoning paradigm which extracts and learns a set of Natural Language rules given the query. The study does not attempt any type of task formalisation into formal logic and does not attempt logic search simulation of any capacity. The core methodologies, goals, and benchmarks of the study are significantly different from those proposed in our study, thus rendering this comparison unfair and uninformative.
> 2. [2] “Language Models can be Logical Solvers” - LogiPit creates a synthetic dataset of a specific solver (PyKe), which contains the query and the heuristically coded/formed iterative implicit fact enrichment process of the solver (In the paper, they create a hand-coded pipeline to have a specific type of an iterative output for the solver). It must be noted that LogiPit does not formalise the query into a program but rather tries to infer unknown/implicit facts given the context. They further use several conversational turns for this formulation instead of direct few-shot execution. The paper further trains a model on the generated dataset and evaluates it on two logical inference tasks. As no code simulation is present, the evaluated models are explicitly trained on data(instead of few-shot eval) and no presence of faithfulness and hallucination detection we consider it unfair and uninformative to compare the paper to our study.
> 3. [3] “Leveraging LLMs for Hypothetical Deduction in Logical Inference: A Neuro-Symbolic Approach” - This paper has been submitted to Arxiv on 29 Oct 2024, thus making it impossible to compare to given the Full Paper Submission Deadline of Oct 1. Also the core methodologies, goals and benchmarks of the study are significantly different compared to the ones proposed in our study, thus rendering this comparison unfair and uninformative.
>
>
> ```Moreover, numerous LLM-based agentic frameworks for reasoning already exist. Consequently, the novelty of this work appears to be limited.```
>
> We respectfully disagree with such an assessment because although LLM-based reasoning frameworks exist, they do not cover solver-free LLM-based search simulation and do not propose methods for measuring faithfulness and detecting hallucinations and sub-optimal reasoning. Consequently, this make the ideas in our paper novel and priorly unexplored.
>
>
>
> ```Clarity: As noted in the initial review, the methodology section remains unclear. The simulating search described in Section 3.1 only covers the generation of a single trace, leaving it unclear how the authors handle path searching and backtracking—arguably the most critical aspects. No detailed pseudo-code is provided to clarify these processes. In contrast, other search-based algorithms, such as Tree of Thought (ToT) and RAP [4], typically include clear pseudo-code in their papers to aid understanding.```
>
> We think there might be a minor misunderstanding here. Section 3.1 covers the complete process of search trace generation. A trace includes all of the paths explored by a model, not a single path. A path can either be successful or lead to failure, in which case we backtrack to the closest sub-goal. As outlined in the benefits of Prolog, the solver completes an exhaustive search over the problem space, which includes several paths and backtracking for failed paths. The in-context samples in LLM demonstrations include traces like that (containing several/all paths and backtracking). We have added details to Section 3 for more clarity.
> Upon the reviewer's request, we have also added a pseud-code for our method in the Appendix A.3

---

> ### Author Response · Authors · 2024-11-24
> **Continuation for the Rebuttal for Reviewer ycsU**
>
> ```The primary baseline used for comparison is F-CoT [7], but there is a scope mismatch. F-CoT primarily targets logical reasoning tasks (e.g., ProntoQA, ProofWriter, FOLIO, LogicalDeduction, AR-LSAT), whereas the benchmarks in this paper focus on math problems and multi-hop QA. It is unclear whether the authors have carefully adapted F-CoT to these benchmarks, raising concerns about the validity of the comparisons.```
>
> We would like to make a clarification here that the “Faithful CoT” that we compare to in the paper is not the [7] (Faithful Logical Reasoning via Symbolic Chain-of-Thought.), which is known/referred to as SymbCoT. Faithful CoT is the fundamental method from the paper “Faithful Chain-of-Thought Reasoning”, which is referred to in the community as Faithful CoT (F-CoT). We had one of the references accidentally misplaced which we have corrected in our revision. We explicitly use all of the benchmarks from the real “Faithful CoT” (“Faithful Chain-of-Thought Reasoning”) study, which does not use logic reasoning benchmarks mentioned by the reviewer. This makes the comparison with Faithful-CoT one-to-one.
>
> Per the reviewers request, we add a direct comparison with to Logic-LM on the hardest sets of ProntoQA[2], AR-LSAT[3] and Logical Deduction [4]. These datasets are used in the original study of Logic-LM and are considered challenging logic inference benchmarks. We opted not to include PrOntoQA and ProofWriter, as they exhibit considerable overlap with the benchmarks already used in our study (e.g., CLUTTER, AQUA, and StrategyQA). Instead, we prioritised datasets that present unique challenges to demonstrate the efficacy of FLARE better.
>
> | Dataset           | Standard | CoT   | Logic-LM       | FLARE          |
> |-------------------|----------|-------|----------------|----------------|
> | **PrOntoQA**      | 47.40    | 67.80 | 61.00          | **73.40**      |
> | **LogicalDeduction** | 40.00    | 42.33 | **65.67**      | 58.60          |
> | **AR-LSAT**       | 20.34    | 17.31 | 26.41          | **27.39**      |
>
>
> Our findings show that FLARE achieves state-of-the-art results on $2$ out of $3$ logic inference benchmarks with $10\%$ increase over CoT and $~7\%$ increase over Logic-LM. These results highlight FLARE’s ability to handle challenging reasoning tasks compared to existing approaches.
>
> ```Comparison to related work: As noted in the initial review, the comparison to related work remains very limited. The proposed method simulates program execution by searching in the problem space, yet it does not include comparisons with classic search algorithms such as Tree of Thought (ToT) and Self-Consistency (SC). Comparing a search-based approach solely with one-pass CoT is insufficient and unfair.```
>
> As mentioned above we directly added comparison to Logic-LM another pioneering study for LLM reasoning. As the Tree of Thoughts and Self-Consistency do not offer ways for measuring faithfulness and do not necessarily produce interpretable answers that follow a given search logic over a defined problem space (Facts, relation, search goal) while being extremely computationally costly, comparing to them would be outside of the scope of the current research and would not add any new revelations.
>
> ```Performance discrepancies: The reported results for Llama-3.1-8B CoT are significantly lower than those presented in the official Llama paper [5] and other studies [6]. For example, its performance on GSM8k is reported as 74.5 in [6] but only 59.2 in this paper; on SVAMP, it is 89.2 in [6] but 58.6 here; and on StrategyQA, it is 68.4 in [6] but 2.9 in this paper. A similar issue is observed with GPT-3.5 CoT. These discrepancies raise concerns about the validity of the evaluation conducted by the authors, leaving the claimed superiority of the proposed method over CoT unclear.```
>
> For CoT results we either directly used the results reported in Faithful CoT(“Faithful Chain-of-Thought Reasoning”) (GPT 3.5) or simply reran the benchmarks using their codebase with prompts, in-context examples without any changes made. This makes sure that we have a 1-1 fair comparison with the original study when we compare FLARE with F-CoT. This would mean that as we either directly report the numbers from the F-CoT paper or run their codebase with only a change of a model that the evaluations are completely valid and the improvement over CoT is fairly measured.

---

> ### Author Response · Authors · 2024-11-24
> **Continuation for the Rebuttal for Reviewer ycsU**
>
> ```Significance: The paper fails to justify the pure prompting-based design, as it essentially renders the approach into yet another fancy CoT or ToT method that could hallucinate during its reasoning.```
>
> We strongly disagree with such a sentiment as our motivations and justifications supported by the results show that the models using FLARE yield an average increase of 23% over CoT and ~60% on F-CoT and yield SOTA results across 7 of the 9 QA tasks that we tested on (original benchmark of Faithful CoT). We have further upon the reviewers request added 3 logic inference benchmarks ProntoQA, AR-LSAT, LogicalDeduction [1,2,3] on which the model shows an average 16% increase over CoT and 8% increase over Logic-LM on 2 out of 3 benchmarks. We show that the increase in results are due to the presence of those simulated code execution by directly comparing performance with plan-only and code approaches (Section 5.2).
>
> The method also allows to explicitly measure the faithfulness of the final answer w.r.t. the performed multi-hop search. Furthermore, FLARE allows direct detection of hallucinations and underutilised knowledge during the search process, which is a unique feature of a reasoning method allowed through code execution simulation. We are able to confirm this way that the optimal searches exhibit, on average, a $18.1\%$ increase in unique emergent facts, a $8.6\%$ higher overlap between code-defined and execution-trace relations, and a $3.6\%$ reduction in unused code relations.
> This means that FLARE proposes novel interpretable reasoning capabilities to LLMs and query exploration and reasoning optimality through simulated code search.
>
> ```Additionally, as noted in the initial review, language models (LMs) are generally more effective at handling natural language than symbolic representations. Integrating symbolic representation into the reasoning process of LMs solely through prompting is likely to degrade their performance, which is shown in [8].```
>
> We would respectfully disagree with such a statement as the use of formalisation for reasoning (through in-context samples or training) has been widely shown to be beneficial during LLM reasoning ([9,10,11]). We would also like to point out that the study [8] ("Let me speak freely? a study on the impact of format restrictions on performance of large language models.") does not in any form hint that reasoning with symbolic representations causes performance degradation. The study [8] only claims that by generating constrained formats such as JSON, YAML etc. the model might face discrepancies. Neither JSON nor a similar format can be compared to a symbolic representation such as a declarative logic programming code, thus rendering the conclusion/analogy/argument mentioned by the reviewer scientifically dubious.
>
> We respectfully request the reviewer to kindly reassess their scores and consider increasing their support for the paper as all of the mentioned concerns, questions, and misunderstandings have been addressed. If there are remaining questions to our work or the rebuttal, we welcome any additional feedback or clarifications.
>
>
> [9] Quan, X., Valentino, M., Dennis, L. A., & Freitas, A. (2024). Verification and Refinement of Natural Language Explanations through LLM-Symbolic Theorem Proving. arXiv preprint arXiv:2405.01379. (EMNLP 2024 Outstanding paper)
>
> [10] Zhou, J. P., Staats, C., Li, W., Szegedy, C., Weinberger, K. Q., & Wu, Y. (2024). Don't Trust: Verify--Grounding LLM Quantitative Reasoning with Autoformalization. arXiv preprint arXiv:2403.18120. (ICLR 2024)
>
> [11] Wang, X., Chen, Y., Yuan, L., Zhang, Y., Li, Y., Peng, H., & Ji, H. (2024). Executable code actions elicit better llm agents. arXiv preprint arXiv:2402.01030.

---

> > ### Comment · Reviewer_ycsU · 2024-11-25
> > **Official Comment by Reviewer ycsU**
> >
> > I appreciate the authors' detailed rebuttals; however, some of my concerns remain unaddressed.
> >
> > ## 1. Novelty
> > > The goal was to create an interpretable method that allows for planning, fuzzy reasoning, and traversing the problem space with backtracking, exact task decomposition, and measuring faithfulness.
> >
> > > We respectfully disagree with such an assessment because although LLM-based reasoning frameworks exist, they do not cover solver-free LLM-based search simulation and do not propose methods for measuring faithfulness and detecting hallucinations and sub-optimal reasoning. Consequently, this make the ideas in our paper novel and priorly unexplored.
> >
> > The proposed method purely relies on prompt engineering with a straightforward design: "plan → code generation → code execution simulation → answer generation."
> >
> > Steps such as planning, path searching, backtracking, and task decomposition have been extensively explored in existing agentic frameworks.
> > [9, 10, 11] (as mentioned by the authors) introduce formalization into reasoning.
> > The simulation of rule/formal logic application with LLMs has been explored in prior work [1,2].
> >
> > All these points make the current research an combination of existing approaches.
> > As a result, the novelty and contribution of this work may not meet the standards typically expected for ICLR.
> >
> > ## 2. Soundness
> > > Section 3.1 covers the complete process of search trace generation. A trace includes all of the paths explored by a model, not a single path. A path can either be successful or lead to failure, in which case we backtrack to the closest sub-goal.
> >
> > How reliable are LLMs at performing path searching in a single pass purely through prompting?
> >
> > [3, 4] suggest that LLMs often struggle with self-discrimination, such as accurately evaluating different reasoning paths. This limitation can lead to self-bias/hallucination, where models favor incorrect or suboptimal paths due to overconfidence or lack of diversity in reasoning.
> >
> > Could the authors compare single-pass path searching with explicit path searching (like Tree of Thoughts)?
> >
> > ## 3. Performance discrepancies
> >
> > > For CoT results we either directly used the results reported in Faithful CoT(“Faithful Chain-of-Thought Reasoning”) (GPT 3.5) or simply reran the benchmarks using their codebase with prompts, in-context examples without any changes made.
> >
> > I don't think this justifies the incorrect experimental results (shown below). Additionally, it doesn't seem appropriate to use the Faithful CoT code repository to evaluate CoT performance. I recommend the authors to re-evaluate the results for CoT.
> >
> > *The reported results for Llama-3.1-8B CoT are significantly lower than those presented in the official Llama paper [5] and other studies [6]. For example, its performance on GSM8k is reported as 74.5 in [6] but only 59.2 in this paper; on SVAMP, it is 89.2 in [6] but 58.6 here; and on StrategyQA, it is 68.4 in [6] but 2.9 in this paper.*
> >
> > In addition, the original Faithful CoT code is designed for GPT-3.5, where it demonstrates high performance. However, the authors directly applied it to other models, such as Llama3.1 and CmDR, without proper adaptation, resulting in 0 accuracy. This comparison is unfair due to the inappropriate experimental setup. I recommend that the authors re-evaluate the results for Faithful CoT after making appropriate adaptations to ensure a fair comparison.
> >
> >
> > ## 4. Comparison to related work
> > > As the Tree of Thoughts and Self-Consistency do not offer ways for measuring faithfulness and do not necessarily produce interpretable answers that follow a given search logic over a defined problem space (Facts, relation, search goal) while being extremely computationally costly, comparing to them would be outside of the scope of the current research and would not add any new revelations.
> >
> > From my understanding, the central claim of the paper is that faithfulness and the incorporation of formal logic enhance reasoning performance.
> > If that is the case, why not compare the proposed method to an approach that excludes these elements (similar to an ablation study)?
> > If all baselines are required to include faithfulness and logic representations, how can the effects of faithfulness and logic representations be effectively investigated?
> >
> > In addition, the agentic framework typically involves substantial token usage.
> > Could the authors provide a comparison of average token usage/accuracy trade-off between the proposed method, CoT, Faithful CoT, Tree of Thoughts and Self-Consistency?

---

> > > ### Comment · Reviewer_ycsU · 2024-11-25
> > > **Continuation for the Official Comment by Reviewer ycsU**
> > >
> > > ## 5. Significance
> > > > We strongly disagree with such a sentiment as our motivations and justifications supported by the results show that the models using FLARE yield an average increase of 23% over CoT and ~60% on F-CoT and yield SOTA results across 7 of the 9 QA tasks that we tested on (original benchmark of Faithful CoT).
> > >
> > > The authors rely heavily on their experimental results to demonstrate the effectiveness of the proposed method. However, the baseline results are limited and questionable (as mentioned above).
> > >
> > > > We would respectfully disagree with such a statement as the use of formalisation for reasoning (through in-context samples or training) has been widely shown to be beneficial during LLM reasoning ([9,10,11]).
> > >
> > > [9, 10, 11] (as mentioned by the authors) all rely on external tools.
> > > In contrast, the primary contribution of the proposed method is to perform formal reasoning without tools.
> > > It is important to note that using external tools and purely prompting LLMs to perform formal reasoning are fundamentally different.
> > >
> > >
> > > ## Reference:
> > > [1] Zhu, Zhaocheng, et al. "Large language models can learn rules." arXiv preprint arXiv:2310.07064 (2023).
> > >
> > > [2] Feng, Jiazhan, et al. "Language models can be logical solvers." arXiv preprint arXiv:2311.06158 (2023).
> > >
> > > [3] Huang, Jie, et al. "Large language models cannot self-correct reasoning yet." arXiv preprint arXiv:2310.01798 (2023).
> > >
> > > [4] Jiang, Dongwei, et al. "Self-[in] correct: Llms struggle with refining self-generated responses." arXiv preprint arXiv:2404.04298 (2024).

---

> > > > ### Author Response · Authors · 2024-11-30
> > > > **Continuation for the Rebuttal for Reviewer ycsU**
> > > >
> > > > ```
> > > > Novelty:
> > > > The proposed method purely relies on prompt engineering with a straightforward design: "plan → code generation → code execution simulation → answer generation."
> > > > Steps such as planning, path searching, backtracking, and task decomposition have been extensively explored in existing agentic frameworks. [9, 10, 11] (as mentioned by the authors) introduce formalization into reasoning. The simulation of rule/formal logic application with LLMs has been explored in prior work [1,2].
> > > > ```
> > > >
> > > > Thank you for highlighting these works! However, we would like to expand/elaborate on the contributions of [1,2] and [9,10,11], and how they relate to this paper.
> > > > 1. [1] “Large Language Models can Learn Rules” - This paper proposes a straightforward LLM prompting inductive-to-deductive reasoning paradigm which extracts and learns a set of Natural Language rules given the query. The study does not attempt any type of task formalisation into formal logic and does not attempt logic search simulation of any capacity. The core methodologies, goals, and benchmarks of the study are significantly different from those proposed in our study, thus rendering this comparison unfair and uninformative.
> > > > 2. [2] “Language Models can be Logical Solvers” - LogiPit creates a synthetic dataset of a specific solver (PyKe), which contains the query and the heuristically coded/formed iterative implicit fact enrichment process of the solver (In the paper, they create a hand-coded pipeline to have a specific type of an iterative output for the solver). It must be noted that LogiPit does not formalise the query into a program but rather tries to infer unknown/implicit facts given the context. They further use several conversational turns for this formulation instead of direct few-shot execution. The paper further trains a model on the generated dataset and evaluates it on two logical inference tasks. As no code simulation is present, the evaluated models are explicitly trained on data(instead of few-shot eval) and there is no presence of faithfulness and hallucination detection. We consider it unfair to compare the paper to our study.
> > > > 3. [9,10,11] - These papers only show the usefulness of formalisation and some of its benefits in LLM reasoning, for explanation generation and other tasks. None of these papers covers the scope of FLARE explained below. (tldr: no code simulation, no faithfulness measurement, no search optimality assessment)
> > > >
> > > > Overall, we obtain significant improvements using FLARE, showing that a solver-free search simulation approach can be used for efficient reasoning. We show the explicit limitations of strictly using external tools and devise methods to assess the faithfulness of the reasoning paths. We further show through our ablations that FLARE allows to explicitly analyse the optimality of completed searches and the explicit reasons leading to correct or incorrect reasoning. Such an approach is completely novel in this line of research. Can the reviewer please elaborate on this issue in light of this?
> > > >
> > > > ```
> > > > All these points make the current research an combination of existing approaches.
> > > > ```
> > > > It is important to note that all research is some combination of prior scientific explorations. However, as mentioned above the research direction within our paper has not been priorly explored by any of the other priorly mentioned works. We obtain significant improvements using FLARE showing that a solver-free search simulation approach can be used for efficient reasoning. We show the explicit limitations of strictly using external tools and devise methods to assess the faithfulness of the reasoning paths. We further show through our ablations that FLARE allows to explicitly analyse the optimality of completed searches and the explicit reasons leading to correct or incorrect reasoning. Such an approach is completely novel in this line of research.

---

> > > > > ### Author Response · Authors · 2024-11-30
> > > > > **Continuation for the Rebuttal for Reviewer ycsU**
> > > > >
> > > > > ```Soundness
> > > > > How reliable are LLMs at performing path searching in a single pass purely through prompting?
> > > > > [3, 4] suggest that LLMs often struggle with self-discrimination, such as accurately evaluating different reasoning paths. This limitation can lead to self-bias/hallucination, where models favor incorrect or suboptimal paths due to overconfidence or lack of diversity in reasoning.
> > > > > Could the authors compare single-pass path searching with explicit path searching (like Tree of Thoughts)?
> > > > > ```
> > > > > We include a complete analysis of the generated paths and the factors that lead to correct and incorrect answers (Section 5.4). We further analyse the paths explicitly for hallucinations, and failures and backtracked paths (Section 5.2-5.4) and their overall impact on the reasoning process. Per the reviewer's request, we have completed another ablation to test for the diversity of the paths. We try to fuzzy match all of the steps (1 hop inferences) in each of the unique paths per search with each other and show that there is no overlap (0.02 average matched hops/inferences per average ~12 hops per search). We also added more explanations and refined these sections, covering the analysis of hallucinations. We show that single-pass pure search simulation does not contain major hallucinations (Table 5-6) that impact reasoning and has diverse paths (multi-hop inferences) per search.
> > > > >
> > > > > ```
> > > > > 3. Performance discrepancies
> > > > > I don't think this justifies the incorrect experimental results (shown below). Additionally, it doesn't seem appropriate to use the Faithful CoT code repository to evaluate CoT performance. I recommend the authors to re-evaluate the results for CoT.
> > > > > The reported results for Llama-3.1-8B CoT are significantly lower than those presented in the official Llama paper [5] and other studies [6]. For example, its performance on GSM8k is reported as 74.5 in [6] but only 59.2 in this paper; on SVAMP, it is 89.2 in [6] but 58.6 here; and on StrategyQA, it is 68.4 in [6] but 2.9 in this paper.
> > > > > ```
> > > > > Thank you for pointing this out we looked further into this and found out that the parsing scheme in the original F-CoT codebase is rater restrictive, thus not allowing for complete recreation of LLama3.1 results. It must be noted that F-CoT uses a different set of instructions, examples and decoding schemes (greedy i.e. temperature=0) that must be used for a for a fair comparison to the current study. As per reviewers request we redid the LLama 3.1-8B experiments with CoT and report the numbers after the improvement/correction of the parsing schemes and other meta parameters.
> > > > > |  | Math Word Problems |  |  |  |  | Multi-hop QA |  |  | Relation |
> > > > > | :---: | :---: | :---: | :---: | :---: | :---: | :---: | :---: | :---: | :---: |
> > > > > | Method | GSM8K | SVAMP | MultiArith | ASDiv | AQuA | StrategyQA | Date | Sport | CLUTRR |
> > > > > | Llama-3.1-8B ${ }_{\text {FLARE }}$ | 72.7 | 86.0 | 96.3 | 83.1 | 62.9 | 70.2 | 59.3 | 76.6 | 36.8 |
> > > > > | Llama-3.1-8B $\mathrm{F}_{\text {-CoT }}$ | 0 | 0 | 0 | 0 | 12.2 | 53.2 | 0 | 0 | 32 |
> > > > > | Llama-3.1-8B ${ }_{\mathrm{CoT}}^*$ | 59.2 | 58.6 | 60.1 | 61.9 | 35 | 2.9 | 20.9 | 95.8 | 42.2 |
> > > > > | LLaMa-3.1-8B-$\mathrm{new}_{\text {CoT }}$ | 85.2 | 82.4 | 91.6 | 79.1 | 51.6 | 43.5 | 74.1 | 89.4 | 45.7 |
> > > > >
> > > > >
> > > > > The new results remove all of the discrepancies pointed out by the reviewer and simulate the LLama3.1-8B results closer to the original report. We can also see that FLARE is able to provide substantial improvement in 5/9 benchmarks. We also checked if the other models are impacted by the priorly mentioned limitations, yet found that the issue is unique to LLama. We added a discussion about this and the improvements in Section 5.1.

---

> ### Author Response · Authors · 2024-11-30
> **Continuation for the Rebuttal for Reviewer ycsU**
>
> ```
> In addition, the original Faithful CoT code is designed for GPT-3.5, where it demonstrates high performance. However, the authors directly applied it to other models, such as Llama3.1 and CmDR, without proper adaptation, resulting in 0 accuracy. This comparison is unfair due to the inappropriate experimental setup. I recommend that the authors re-evaluate the results for Faithful CoT after making appropriate adaptations to ensure a fair comparison.
> ```
> The Faithful CoT paper does not mention explicit limitations of the model use. Our results show the brittlness of code generation with models that are not explicitly tuned for coding or fail to produce code in a strict parsable format. This shows the explicit limitation of approaches like F-CoT and Logic-LM that if the model fails to produce executable or explicitly formatted output than the execution of the generated code becomes impossible and massively degenerates the results. FLARE is not dependent upon such limitations which is a clear advantage for a solver free LLM based method.
>
> We would like to ask for clarifications regarding the proper adaptations mentioned by the reviwer as we find no natural changes that can be made to the F-CoT approach for adapting to other models. Per the reviewer's request, we explicitly tested a change in instructions and the formatting of the in-context samples, yet the F-CoT results still remain consistent. We consider this a fair comparison as we 1-1 produce the results of the F-CoT with the suggested hyperparameters, instructions and in-context samples.
>
> ```
> Comparison to related work:
> From my understanding, the central claim of the paper is that faithfulness and the incorporation of formal logic enhance reasoning performance. If that is the case, why not compare the proposed method to an approach that excludes these elements (similar to an ablation study)? If all baselines are required to include faithfulness and logic representations, how can the effects of faithfulness and logic representations be effectively investigated?
>
> In addition, the agentic framework typically involves substantial token usage. Could the authors provide a comparison of average token usage/accuracy trade-off between the proposed method, CoT, Faithful CoT, Tree of Thoughts and Self-Consistency?
> ```
> It must be noted that in order to produce ToT or Self-Consistency results, we must set the decoding temperature to a number bigger than zero, i.e. greedy decoding is not possible (temperature usually >= 1). However, in our study, we used greedy decoding (for CoT, F-CoT and FLARE) for complete reproducibility and fair comparison. Nonetheless, per the reviewer's request, given the tight timeframe, we produced a direct comparison with ToT on two challenging logical inference datasets. We will submit the completed benchmark with ToT for the final version of the paper.
>
> | Dataset  | ChatGPT (gpt-3.5-turbo) |       |          |                   |           |              |
> |----------|-------------------------|-------|----------|-------------------|-----------|--------------|
> |          | Standard                | CoT   | Logic-LM | ToT_{DFS_depth=3} | FLARE     | FLARE_{SR=2} |
> | PrOntoQA | 47.40                   | 67.80 | 61.00    | 68.67             | **73.40** | **79.4**     |
> | AR-LSAT  | 20.34                   | 17.31 | 26.41    | 26.92             | **27.39** | **30.73**    |
>
>
>
> Our results show that using FLARE allows to achieve significant improvements over ToT.

---

> > ### Author Response · Authors · 2024-11-30
> > **Continuation for the Rebuttal for Reviewer ycsU**
> >
> > The speed of the external solvers used for baseline methods (F-CoT) is highly dependent and sensitive towards the executing system and choice of the setup that runs it (F-CoT can be run with different versions of Python, PDDL etc). This makes a direct comparison with LLM inference impossible. However, per the reviewer's request, we compare the number of tokens and the inference speed for generating CoT vs F-CoT (without the external solver added time) vs FLARE.
> >
> > | model          |   ('plan', 'mean') |   ('plan', 'std') |   ('code', 'mean') |   ('code', 'std') |   ('trace', 'mean') |   ('trace', 'std') |   ('trace_percentage', 'mean') |   ('trace_percentage', 'std') |
> > |:---------------|-------------------:|------------------:|-------------------:|------------------:|--------------------:|-------------------:|-------------------------------:|------------------------------:|
> > | Llama-3.1-8B   |            552.544 |           146.769 |            241.274 |           128.231 |             336.255 |            281.753 |                        26.9211 |                       11.3238 |
> > | command-r      |            511.435 |           160.666 |            180.228 |           132.454 |             266.82  |            268.343 |                        23.0468 |                       15.1715 |
> > | command-r-plus |            577.032 |           184.77  |            189.203 |           151.211 |             223.105 |            209.69  |                        19.7404 |                       12.5113 |
> > | gpt-3.5-turbo  |            557.774 |           161.733 |            177.533 |           128.255 |             164.41  |            184.529 |                        16.3258 |                       10.686  |
> >
> >
> > Our results show that the total length of the trace is, on average, composed of ~250 tokens which corresponds to an approximately 20% increase compared to CoT and F-CoT. Despite this marginal increase, the impact on overall inference speed is negligible w.r.t. to Cot and F-CoT. In particular, the additional tokens are not a reason for a computational overhead as the computational cost of adding a small number of tokens (such as the additional ~250 tokens in FLARE traces) is minimal compared to the overall cost of processing the base sequence. This is specifically true when the sequence lengths remain well below the model's maximum context window.
> >
> > ```
> > 5. Significance
> > The authors rely heavily on their experimental results to demonstrate the effectiveness of the proposed method. However, the baseline results are limited and questionable (as mentioned above).
> > ```
> > We have addressed this comment when discussing/addressing “3. Performance discrepancies”.
> >
> > We hope that this rebuttal has cleared all the misunderstandings and provided all the clarifications requested by the reviewer.
> >
> > We respectfully request the reviewer to kindly reassess their scores and consider increasing their support for the paper as all of the mentioned concerns, questions, and misunderstandings have been addressed. If there are remaining questions to our work or the rebuttal, we welcome any additional feedback or clarifications.

---

> > > ### Author Response · Authors · 2024-12-01
> > >
> > > Thank you for your valuable feedback on our submission. We’ve addressed your comments and provided detailed clarifications in our rebuttal. We hope that the clarifications provided and the questions addressed in our rebuttal have resolved any concerns, allowing the reviewer to consider increasing the support for the paper.

---

> > > ### Comment · Reviewer_ycsU · 2024-12-03
> > > **Official Comment by Reviewer ycsU**
> > >
> > > Thanks for the authors' detailed response.
> > >
> > > The introduction of external solvers aims to provide a more reliable, though narrower, mechanism to support LLMs. FLARE bypasses this process by relying purely on prompting, which is generally considered less effective due to insufficient training data on formal logic.
> > >
> > > The comparison with CoT maybe not entirely fair. FLARE is essentially an agentic framework augmented with formal logic. To accurately assess its effectiveness, it would be more appropriate to compare it against other agentic frameworks without formal logic. The reason is that agentic frameworks are typically better than CoT (particularly for weaker LLMs). The observation that FLARE sometimes outperforms CoT (based on the corrected results) raises questions about the source of its benefits. It remains possible that a similar agentic framework without formal logic could outperform FLARE.

---

> ### Author Response · Authors · 2024-12-03
> **Continuation for the Rebuttal for Reviewer ycsU**
>
> We thank the reviewer for their comments.
>
> > The comparison with CoT maybe not entirely fair. FLARE is essentially an agentic framework augmented with formal logic. To accurately assess its effectiveness, it would be more appropriate to compare it against other agentic frameworks without formal logic. The reason is that agentic frameworks are typically better than CoT (particularly for weaker LLMs).
>
>  What kind of experiments would the reviewer like to see?  Does the reviewer have any references for agentic frameworks we can include that have not already been included in our comparisons?
>
>  We followed the evaluation protocols in, e.g. the F-CoT paper (https://arxiv.org/abs/2301.13379; where authors compare with vanilla CoT and another prompting-based baseline) and added more baselines (Logic-LM and ToT per the reviewers request), but we are happy to include more comparisons if that helps the quality of the paper!
>
> We, as mentioned priorly, also include a rigorous analysis of our framework compared to CoT and F-CoT and explicitly highlight the strengths of the proposed method (Section 5.2-5.4).

---

### Meta-Review · Area_Chair_PNEt · 2024-12-19

**Metareview:**

This work investigates a prompting workflow for solving reasoning tasks where a problem is formalized as a logic program but its execution is simulated by a language model rather than delegated to an external solver, which theoretically trades precision for coverage, and allows quantifying the extent to which the language model is faithfully simulating the solver. An advantage of this is that it allows a more careful experimental study of reasoning breakdowns: Does reasoning fail due to failures of formalization or failures of execution? It also leads to modest gains across benchmarks, at least against the baselines considered. The weaknesses of the paper are that it is effectively a prompting pipeline, and it foregrounds the engineering results instead of the scientific insight that might be obtained by experimental study of reasoning breakdowns, as well as problems with the baselines described below.

It is particularly unclear that the baselines are fair: The authors are unable to reproduce strong numbers from various baselines, and instead present lower numbers that they themselves generated. *This might not be the author's fault!* Perhaps the original papers got it wrong. But it should give pause, especially been combined with the fact that *their reproduction of F-CoT does not even do the minimal check of filtering programs for syntactic correctness*, leading this baseline to have a misleading "0%" accuracy. Given that they extend the same grace to their own method (from the rebuttal), it is hard to have a charitable interpretation of the baseline numbers.

The biggest reason to reject the paper is because of these issues with the baselines. There are other reasons as well, such as the fact that it is essentially a prompting pipeline with an emphasis on the engineering result instead of scientific insight, and that the engineering itself is not particularly methodologically novel.

**Additional Comments On Reviewer Discussion:**

ycsU raised valuable insights about the problems with the baselines, which were not effectively rebutted: In fact, it came out that they were extending grace to their own method which was not granted to the baselines, and experiments requested by ycsU exposed narrowness in the empirical advantage of the method. Although there is a champion, bKpGf, they also concede that there are weaknesses in the evaluation.

---

### Decision · Program_Chairs · 2025-01-22

Reject